# Characterizing the Ventral Visual Stream with Response-Optimized Neural Encoding Models

**Meenakshi Khosla[1,*], Keith Jamison[3], Amy Kuceyeski[3,4] and Mert R. Sabuncu[1,2,3]**

[1] Department of Brain and Cognitive Sciences, Massachusetts Institute of Technology, Cambridge, MA, 02139
[2] Nancy E. and Peter C. Meinig School of Biomedical Engineering, Cornell University, Ithaca, NY 14853
[3] Radiology, Weill Cornell Medicine, New York, NY 10065
[4] Brain and Mind Research Institute, Weill Cornell Medicine, New York, NY 10065
[*] Correspondence: mkhosla@mit.edu

## Abstract

Decades of experimental research based on simple, abstract stimuli has revealed the coding principles of the ventral visual processing hierarchy, from the presence of edge detectors in the primary visual cortex to the selectivity for complex visual categories in the anterior ventral stream. However, these studies are, by construction, constrained by their *a priori* hypotheses. Furthermore, beyond the early stages, precise neuronal tuning properties and representational transformations along the ventral visual pathway remain poorly understood. In this work, we propose to employ response-optimized encoding models trained solely to predict the functional MRI activation, in order to gain insights into the tuning properties and representational transformations in the series of areas along the ventral visual pathway. We demonstrate the strong generalization abilities of these models on artificial stimuli and novel datasets. Intriguingly, we find that response-optimized models trained towards the ventral-occipital and lateral-occipital areas, but not early visual areas, can recapitulate complex visual behaviors like object categorization and perceived image-similarity in humans. We further probe the trained networks to reveal representational biases in different visual areas and generate experimentally testable hypotheses.

## 1 Introduction

Understanding the nature of representations and computations in the visual system has been a longstanding goal in neuroscience. Substantial progress been made by presenting model organisms with simple, abstract stimuli like noise or sine-wave gratings and studying the evoked responses. These carefully crafted experiments, for instance, revealed the presence of edge detectors in the primary visual cortex [1] and sparked the search for the 'optimal input' beyond early visual areas across the visual cortical hierarchy. Significant efforts have also been made to understand the feature complexity of tuning in V4 and IT neurons [2, 3, 4, 5]. Over the last two decades, through numerous experiments, recording and analysis techniques, a conceptual understanding has emerged that early visual areas extract low-level features like edges or curves, mid-level regions extract complex local shapes, and high-level regions encode semantic categories, like faces or scenes [6, 7, 8, 9, 10]. While much of the details remain unknown, there is accumulating evidence that visual categorization occurs in the ventral temporal cortex in humans [11]. This understanding was enabled by synthesizing findings across different hypothesis-driven studies, most of them employing artificial stimuli atypical of real-world vision, thus lacking ethological validity.

Recently, deep convolutional neural networks (CNNs) optimized for computer vision (CV) tasks, have emerged as the most accurate models of the primate ventral visual stream [12, 13, 14, 15]. Furthermore, these CV task-optimized models offer a hierarchical correspondence to visual areas along the ventral stream: early CNN layers best predict V1, while intermediate and late layers best predict V4 and IT [12, 16, 17, 18, 19, 20, 21, 22]. They provide intelligibility in the form of the goal-driven perspective, highlighting the functional role of representations [23]. However, we are far from a complete understanding of the top-down computational objectives (the battery of ethologically-relevant tasks) that may have shaped neural representations through evolution or experience-based learning; in the absence of right hypotheses, the task-optimized approach can fall short. Studies have already revealed significant inconsistencies between these models and human behavior and gaps remain in their respective robustness to distortions, image-level consistency etc [24]. Recently, the idea of using neural data in the model development process has gained some traction, and demonstrated success in increasing both model-human behavior [25, 26, 27], and model-brain alignment, while revealing novel insights about information processing in neural systems [28, 29, 30, 31, 32].

In this study, we leverage recent advances in large-scale fMRI data collection [33] to train hypothesis-agnostic computational models directly on neural data with novel predictive precision. Importantly, the stimulus set employed in this study contains crowded images of multiple objects in their natural contexts, thus being more typical of everyday scenes and allowing us to characterize neural representations and computations in rich, naturalistic conditions. We demonstrate the remarkable ability of these "response-optimized" neural encoding models in reproducing the stimulus-response relationship in voxels along the ventral visual stream for arbitrary images, equaling or surpassing the unprecedented prediction performance of task-optimized models. We show that these models generalize well to contexts that are vastly different from their training conditions (artificial stimuli, novel subjects, etc.), providing a strong quantitative validation for the response-optimization approach of building encoding models for neurobiological visual systems beyond the retina [34]. Our approach further goes beyond prediction accuracy and aims to gain insights about neuronal tuning properties. Specifically, we implement neural encoding models that disentangle the processing of the "what" and the "where." This allows us to reveal the selectivity of individual brain voxels along both location and appearance dimensions. We first demonstrate that the estimated spatial topographic organization from our neural encoding models exhibits remarkable agreement with the fine-scaled retinotopic organization of the cortex, which is typically measured with dedicated, painstaking experimental paradigms. We analyze the features that emerge in the neural encoding models to understand the tuning properties of cortical voxels. This analysis suggests shape-biased tuning across the ventral visual stream, increasing in complexity from Gabor-like features in V1, curves/angles in V2 to concentric and complex shape processing in V4 and higher-order lateral-occipital and ventral-occipital ROIs respectively. We also go beyond single-voxel analysis to characterize how *representations* evolve along the ventral visual pathway. Probing the information encoded in model-predicted patterns of activity reveals the monotonically increasing separability of category information along the ventral stream hierarchy, strongly supporting the computational goal of the ventral visual stream as a visual categorization system. We further demonstrate the correspondence between model representations (of high-level visual areas) and mental representations (as assessed with human similarity judgements), suggesting that these response-optimized models inherently capture human psychological judgements, without any explicit supervision to do so. Furthermore, a principle components analysis reveals that models of high-level visual ROIs capture complex stimulus dimensions like faces, bodies and shapes. Our paper illustrates how neural encoding models trained on hypothesis-agnostic, naturalistic datasets can be used to confirm neural phenomena characterized painstakingly using controlled experimental paradigms that might be hard to replicate in individual subjects. More importantly, however, we show how *in silico experiments* on neural encoding models can generate novel hypotheses and provide a powerful framework for building broad, generalizable theories about neural information processing.

## 2   Materials and Methods

**Natural Scenes Dataset**    A detailed description of the Natural Scenes Dataset (NSD [1]) is provided elsewhere [33] (see also the Appendix). The dataset contains measurements of fMRI responses from 8 participants who each viewed 9,000–10,000 natural scenes. A special set of 1,000 images were

---

[1]http://naturalscenesdataset.org

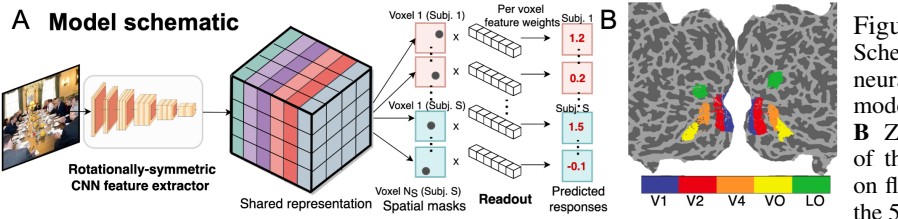

**A  Model schematic**

Rotationally-symmetric CNN feature extractor

Shared representation · Voxel N_S (Subj. S) Spatial masks · Readout · Predicted responses

Voxel 1 (Subj. 1) · Per voxel feature weights · Subj. 1 — 1.2 · 0.2

Voxel 1 (Subj. S) · Subj. S — 1.5 · -0.1

**B**

V1  V2  V4  VO  LO

Figure 1: **A** Schematic of the neural encoding model architecture. **B** Zoomed-in view of the visual cortex on flatmap, showing the 5 visual ROIs.

shared across subjects; the remaining were mutually exclusive. We focused on modeling responses within 5 visual cortical ROIs in the study. Three ROIs belonging to the retinoptic early visual cortex, namely, V1v, V2v and hV4 were defined using a population receptive field (pRF) localizer scan session [33]. Two higher order ROIs, namely ventral-occipital areas (VO1-2) and lateral-occipital areas (LO1-2) were delineated using a visual probabilistic atlas [35](Figure 1B).

**Response-optimized neural encoding model**    We trained separate voxel-level neural encoding models for each of the above regions with the same backbone architecture. Keeping the architecture constant across visual areas ensured that differences in emergent representations are driven mainly by the nature of response patterns. The predictive model comprises a convolutional neural network (CNN) *feature extractor* shared across all subjects, but unique to specific visual areas. We employ a linear *readout* model on top of the feature space to predict the responses of individual voxels in a specific region. The linear readout is *factorized* into *spatial* and *feature* dimensions following [36] (Figure 1A). This allows us to separate spatial tuning (the "where") from feature tuning (the "what"). The shared CNN feature extractor consists of four convolutional blocks, with each block comprising the following feedforward computations: two $3 \times 3$ convolutional layers, each followed by an inner batch norm, and a ReLU; and an anti-aliased AvgPool operation (stride = 2) at the end. We employ E(2)-steerable convolutions in all our models to extract features invariant to orientation [37, 38]. This modeling choice is inspired by neural computations in early visual areas where groups of neurons are known to perform similar computations, e.g., edge or curve detection, at different orientations. Each convolutional layer contains 48 feature sets extracted at 8 orientations. Weights of the readout are a computed as outer products between a spatial filter and a feature vector. The spatial filter further has a positivity constraint (enforced using rectification) and is length-normalized for each voxel.

**Training and testing response-optimized encoding models**    We used 4 NSD subjects for training and reserved the other 4 subjects for studying generalization. The first group collectively saw 37,000 natural scene images, including the 1,000 shared images. We used the 1,000 shared images for testing our models and split the remaining stimulus set into 35,000 training and 2,000 validation images. All parameters of the neural encoding model were optimized jointly to minimize the masked *mean squared error* between the predicted and measured fMRI response. This loss allows us to propagate errors through the shared CNN even if the subjects are not exposed to common stimuli since we can exclude the subjects/voxels for which the target response is not present from the loss calculation. We quantify 'predictive accuracy' on the test images as the Pearson's correlation coefficient between the predicted and measured fMRI response at each voxel. We also compare this against the noise-ceiling, expressed as the signal-to-noise ratio when considering the response variability of each voxel across 3 repetitions per image (see Appendix A.5). Table A.1 provides a summary of all analyses methods and baselines used for evaluating and interpreting the proposed response-optimized models.

**Baseline models**    **(a) Category ideal observer model:** We fit a non-image-computable model using human-generated annotations for NSD images (obtained from the MS-COCO database). The input to the categorical model is a k-hot encoding vector corresponding to the 80 object categories annotated in the database, where each element indicates the presence/absence of the corresponding category in the image. We fitted $l_2$ regularized linear regression models on this space to predict voxel responses for held-out stimuli (Appendix A.8). **(b) Task-optimized Models:** We investigated several DNN models, including AlexNet [39], ResNet-50 [40], DenseNet [41] and CORnet-s [42], optimized for object recognition on the large-scale ImageNet dataset [43] (Appendix A.3, Table A.2). Since these models can demonstrate lower prediction performance due to the domain shift between ImageNet and NSD stimuli, we also employed a ResNet-50 architecture pre-trained for object detection on the MS-COCO dataset [44] as an additional baseline (since NSD stimuli were

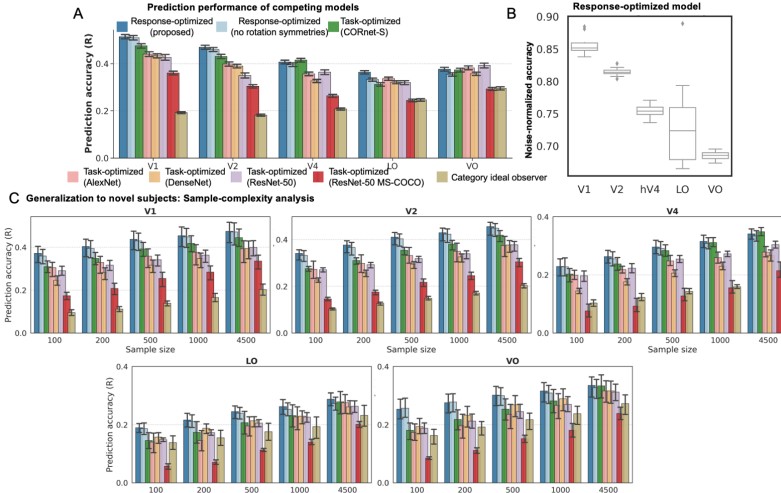

Figure 2: **Quantitative analysis.** **A** Prediction performance of competing models across the 5 visual areas. **B** Boxplot depicting the noise-normalizing predictivity of the proposed response-optimized model. **C** Generalization of competing models to novel subjects as a function of sample size used for fitting their respective response functions (or readouts). Error bars in barplots depict the 95% confidence-interval (CI) around estimated mean.

drawn from the MS-COCO dataset). For CORnet-S, we pre-select the layer homologous to each visual region and employ a *spatial × feature* factorized readout (same as the response-optimized model) on top of the fixed feature backbone, allowing a more direct comparison to the proposed model. For the remaining models, we followed the more standard procedure for fitting task-optimized models to voxel responses based on a full linear readout. We extracted features from pre-trained layers of each network and fitted $l_2$ regularized linear regression models separately for each layer. The regularization parameter was optimized independently for each subject, each layer, each model and for voxels in each visual area by testing among 8 log-spaced values in [1e-4, 1e4]. For each visual area and model, we identified the layer that best predicts responses on the validation set. These model layers were used to quantify test performance. **(c) Response-optimized (no rotation symmetries):** Here, we train response-optimized models with the same architecture as the proposed models but without sharing weights across filter orientations.

**Characterizing neural representations by behavioral comparisons:** **(a) Assessing separability of category and semantic information along the visual hierarchy** To probe the learned featural tuning in the response-optimized model for each visual area, we used representational similarity analysis (RSA) [45]. Here, we correlated the patterns of predicted activity between multiple exemplars from different categories of stimuli to obtain a representational similarity matrix (RSM) for each visual area. Further, we quantified the separability of categorical information within these similarity matrices by computing a correlation coefficient (Kendall's $\tau$) between model RSMs against a ground truth adjacency matrix defined by object category labels. Specifically, the elements of this matrix are 1 if the corresponding two images belong to different categories and 0 if the images belong to the same category. This analysis was first performed using a subset of 104 images from the THINGS database [46] that belong to a pre-defined set of categories for ease of visualization. These categories were chosen in accordance with previous fMRI studies employing RSA [47, 48] and include the following: {face, hand, elephant, cat, plant, fruit, car, tool}. Subsequently, we also validated the obtained results using another large-scale vision dataset, namely ImageNet-16 [49]. We sampled 500 stimuli from each ImageNet-16 category, resulting in a total of 8,000 images and computed the similarity between model RSMs and the categorical adjacency matrix following the same approach described for the THINGS dataset. To assess visual information flow in relation to high-level semantic understanding, we computed pairwise similarity between predicted responses for held-out NSD images to construct model-predicted RSMs for each visual area. We constructed a semantic RSM by correlating the 80-D binary category vectors (which indicate whether each category was present in the image or not) for every pair of these NSD images and quantified agreement with model-predicted RSMs using Kendall's $\tau$ (henceforth, termed the 'NSD: Semantic' analysis). We computed this agreement using *measured* RSMs for all visual areas as well. In RSA described here and anywhere else, error bars (95% CI) are computed by performing bootstrap sampling over voxels.

**(b) Assessing alignment with human perception** We compared model representations against behavioral data from a Multiple Arrangement task where participants interpreted 100 NSD images and arranged them according to the similarity of their content (this dataset called 'nsdmeadows' was released with NSD [33]). Critically, these images were not seen by any of the response-optimized models. We constructed a model Representational Dissimilarity Matrix (RDM) for each visual ROI using the correlation distance metric for quantifying pairwise dissimilarities between model-predicted voxel responses. We compared model RDMs against human pairwise dissimilarity judgements using the Kendall's $\tau$ coefficient and call this the 'NSD: perception' analysis.

## 3 Results

**Response-optimized models approach the noise ceiling for prediction performance and generalize to novel subjects with low sample complexity** We observed a remarkably high noise-normalized prediction accuracy ($\sim$70-85%) using the proposed rotationally-symmetric response-optimized model in all 5 visual ROIs, including the higher-order ROIs (LO1-2 and VO1-2) (Figure 2B, Appendix Figure A.2, A.6). Next, we investigated the possible quantitative advantages of learning a rotationally-symmetric feature space against the representations from a standard convolutional architecture with no weight sharing across orientations. Importantly, we observe that the proposed rotationally-symmetric convolutional architecture performs on par with a standard CNN (Figure 2A), even outperforming the latter in certain visual ROIs, despite containing fewer parameters. This suggests that sharing weights across filter orientations provides a strong inductive bias, allowing us to fit expressive models efficiently. We further compared these models against ideal observer and task-optimized models, and found that response-optimized models consistently outperform these baselines in early visual areas. In high-level visual areas, the response-optimized model significantly outperforms the ideal observer model, while performing on par with or even surpassing some task-optimized models.

Next, we assessed how each of these predictive models generalizes to the remaining set of 4 test subjects. This analysis intended to compare the statistical efficiency afforded by different representations [50]. As such, we examined the trend of how much stimulus-response data the models need to achieve a certain level of prediction performance [50]. Here, we freeze the shared feature extractor network and vary the amount of stimulus-response pairs from the 4 new subjects to train the linear readouts (Appendix A.7). We do the same for the task-optimized baselines, where the linear prediction model is fit on new subjects. As shown in Figure 2, the response-optimized models can achieve a significantly high accuracy even when only just 5% of the training examples from novel subjects ($\sim$ 500 samples) are used to train the readout. This remarkable generalization of response-optimized networks, exceeding that of task-optimized networks, to novel subjects suggests our models contain appropriate "inductive biases" that improve sample efficiency.

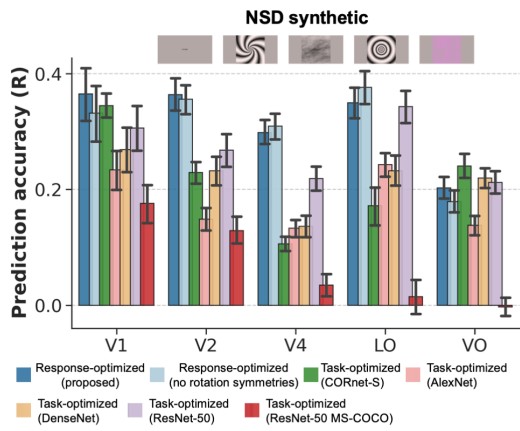

Figure 3: **Generalization to synthetic stimuli.** OOD generalization of competing models to synthetic (non-naturalistic) stimuli from *nsdsynthetic* experiment. Sample stimuli are shown on the top. Error bars depict the 95% CI around estimated mean.

**Computational models generalize to OOD datasets** We next investigated out-of-distribution (OOD) test settings to understand the generalization behavior of response-optimized models. The first OOD setting we consider is the NSD-synthetic experiment, comprising fMRI recordings to synthetic stimuli (samples shown in Figure 3) from the same NSD subjects. As shown in Figure 3, response-optimized models significantly outperform task-optimized models (except in region VO) in this OOD domain ('artificial' stimuli), suggesting that the mapping from task-optimized DNN features to voxel responses, was domain/dataset-specific. Response-optimized models, on the other hand, allow for flexible OOD generalization. Generalization to other novel fMRI datasets, including [51, 52, 53] is shown and discussed in the Appendix B.2.

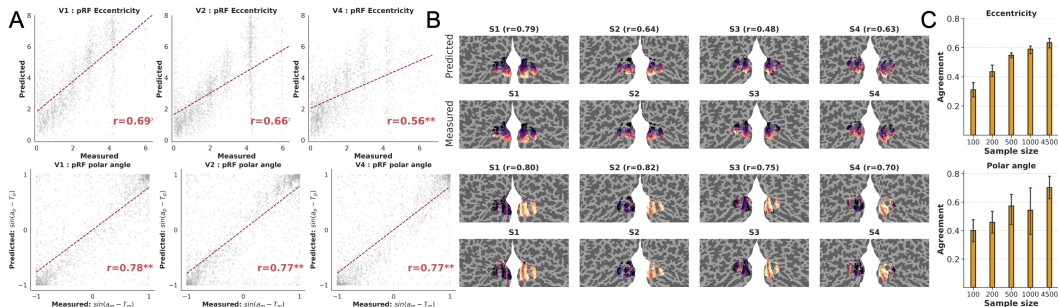

Figure 4: **pRF estimation with response-optimized encoding models**. **A** Scatter plots showing predicted and localizer-estimated retinotopic parameters for all voxels in all 4 subjects (see Suppl. Methods for polar angle metrics shown on scatter plots). Inset correlations are computed using voxels from all 4 subjects. **B** Predicted and localizer-estimated retinotopic parameters visualized on cortical flatmaps. Bolded values on top show subject-specific agreement. **C** Agreement between estimated and measured retinotopic maps as a function of training examples (stimulus-response pairs) from novel subjects used to train their linear readout. Error bars depict the 95% CI around estimated mean.

**Learned spatial masks reproduce the fine-grained retinotopic organization of early visual cortex**
The factorized readout in the proposed model allows us to separate spatial and feature selectivity. We characterized the spatial tuning of early visual areas by estimating the population receptive field (pRF) from the learned spatial masks of the proposed response-optimized model; we compare the estimated parameters against measurements from the independent NSD pRF localizer experiment that uses artificial stimuli. The pRF is defined as the region of the visual field within which a stimulus results in increased aggregated activity across a population of neurons, as reflected in fMRI. Based on the analysis of these data, each brain voxel's retinotopic organization can be described with three parameters: the pRF location in polar coordinates (polar angle and eccentricity), and size. Using the voxel-wise spatial mask of the readout in our encoding model, we estimated the pRF location as the average of the coordinates weighted by the learned spatial mask value at each grid location. To compare the estimated eccentricity against the corresponding measurements from pRF localizers, henceforth called *'eccentricity agreement'*, we calculated the Pearson's correlation coefficient between the measured and predicted eccentricity arrays in each ROI. To compare the estimated polar angle values against corresponding measurements from pRF localizer scans, henceforth called *'polar angle agreement'*, we adopted a measure from circular statistics called circular correlation coefficient [54], due to the circular nature of angular data (Appendix A.4). It is important to note that the encoding models do not account for the spatial locations of target voxels. Despite this, the estimated spatial topographic organization from the encoding models exhibits remarkable levels of agreement with the fine-scaled retinotopic organization of the cortex without any explicit supervision to do so (Figure 4A,B): *eccentricity agreement* $\in$ [0.48-0.79] for all 4 subjects whereas the *polar angle agreement* $\in$ [0.70-0.82]. We further compared this agreement on the estimated pRF parameters for the held-out set of 4 NSD subjects by varying the quantity of data used to estimate their linear readouts from the fixed convolutional core of the response-optimized model (pre-trained on the initial set of 4 subjects). We find that while the agreement increases with the sample size used to fit readouts, even with much fewer samples (e.g. 100), we can still characterize the retinotopic organization (Figure 4C). This is better than the retinotopic agreement observed for the CORnet-S model (which also employed a factorized readout, enabling characterization of spatial tuning (Appendix Figure A.8)). Previously, retinotopic organization of the human visual cortex has been dominantly studied with spatially modulated stimuli. While encoding models have been shown to be broadly useful for characterizing retinotopy [55, 31], a precise quantification of the agreement between model-estimated and localizer-estimated retinotopic parameters, as well as the effect of sample sizes on this agreement, has been lacking. Here, we present an alternate approach based on naturalistic stimulation and response-optimized encoding models to delineate subject-specific retinotopic maps and precisely quantify the agreement. Remarkably, the retintopic organization estimated from the proposed encoding models is even more spatially smooth than the localizer-based organization, despite not being informed by anatomy.

**Response-optimized models characterize transformations of image representations along the visual cortical hierarchy**

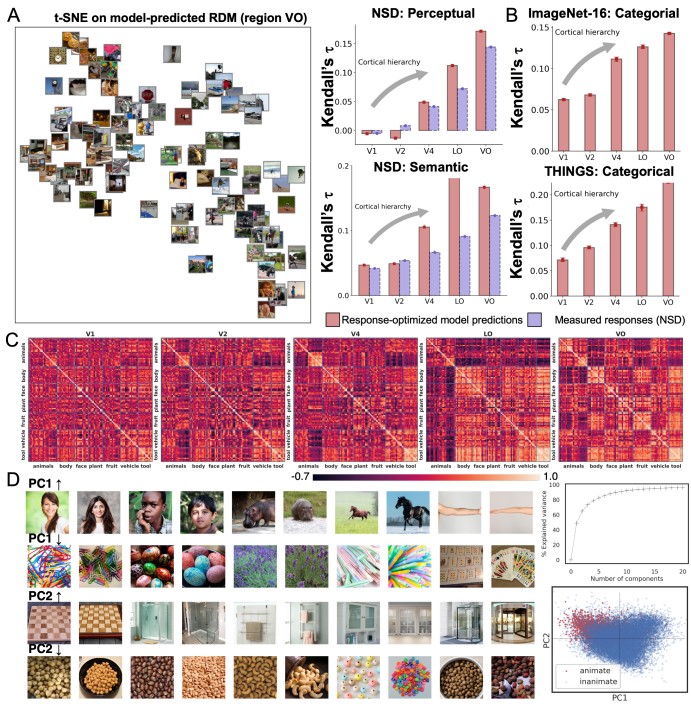

Figure 5: **A** highlights that models of high-level visual areas (LO and VO) are more aligned with human perception and image semantics than early visual ROI (V1-V4) models. **B** and **C** highlight the **separability of category information** along the ventral visual stream. **B** Results of the Representational Similarity Analysis (RSA) framework applied to the THINGS and ImageNet-16 datasets containing different categories of objects. **C** Matrices of all pairwise similarities between model-predicted responses for sampled images in the THINGS dataset. **D** [Left] Top 2 images for the 5 most and least activating concepts of the first two PCs of model-predicted **VO** responses. [Upper right] Total explained variance as a function of the number of PCs. [Lower right] All images from the THINGS dataset projected onto the first two principal dimensions of the response and colored by their labels.

**(a) Monotonically increasing alignment with human perception along the ventral visual stream** We evaluated models on behavioral data from a Multiple Arrangement task where participants arranged NSD images according to the similarity of their content. This behavioral analysis of neural network models (Figure 5A) suggests that higher visual areas (LO and VO) more strongly capture human perceptual patterns and image semantics than early visual areas and this quantitative agreement increases monotonically along the ventral visual hierarchy ('NSD perception' and 'NSD semantics' analysis discussed in Methods). Importantly, responses patterns in their respective models group semantically similar categories together, as shown in the t-SNE plot for model-predicted RDM (region VO). For instance, faces are all clustered in the bottom right of the t-SNE plot and scenes group together in the left (Figure 5A), despite wide variations in the visual appearance of these categories. This is remarkable, given that the models received no supervision from perceptual/semantic measures. Human behavioral data ('nsdmeadows') appears very similar and is shown in Appendix Figure A.10. This suggests that these models, solely optimized for high-level visual cortical response prediction – without any semantic supervision – can even serve as powerful models for uncovering the structure of mental representations. Intriguingly, the representational structure of predicted responses exhibits stronger agreement with perceived similarity than the representational structure of measured responses. This suggests that both the perceptual and semantic information are likely more explicit in predicted responses (even for held-out data) than measured responses. One possible explanation is that fMRI measures of brain activity have undesired noise components and predictive models, in particular, response-optimized models as shown here, can *denoise* this fMRI activity (see also Appendix Figure A.3).

**(b) Monotonically increasing separability of category information along the ventral visual stream** Task-optimized computational models have suggested that the ventral stream hierarchy 'untangles' representations through a sequence of processing steps, so that representations of objects and categories that are inseparable at the initial processing stage (e.g., V1) become untangled at the last stage of the hierarchy [56, 23]. Here, we present a strong evidence for this increased separability through hypothesis-agnostic response-optimized models. Previous studies have shown that the representational geometry for the same set of stimuli varies systematically across different cortical areas, providing a useful signature for how information is transformed across different cortical processing streams [56]. Here, we employ the representational similarity analysis (RSA) framework to relate emergent representations in response-optimized models of difference brain regions against a ground truth adjacency matrix defined by object category labels. Keeping model

architecture the same, simply by changing the response targets for an encoding model from voxels in visual areas V1 through LO-VO, we observe a drastic change in the geometry of the extracted representation. The increasingly prominent block-diagonal structure in the RSMs along the ventral visual stream (Figure 5C), consistent with categorical distinctions in the THINGS dataset, highlights that the distributed activity patterns to exemplars of the same category become increasingly more similar; and patterns to exemplars of different categories become progressively dissimilar along the processing stream, strongly supporting the computational goal of the ventral visual stream as a categorization system. We also quantified the separability of categorical information in different visual areas (Figure 5B). Here again, we see that this finding regarding increasing separability of categorical information holds across both datasets containing different visual categories.

**Response-optimized models reveal high-level visual information within ventral-occipital cortex**
Next, we wanted to discover what kinds of features emerge in the ventral-occipital region model and whether they are capable of discriminating between complex categories. Instead of looking at all model neurons emulating voxels, we built a compact neural representational space by passing a large stimulus set (∼27,000 images from the THINGS database [46]) through the predictive models and then performing Principal Components Analysis (PCA) on the predicted responses of this network (Appendix A.6). Visualizing the images from top concept categories that elicit highest and lowest activation of individual principal components (PCs) reveals complex stimulus dimensions: the first PC, which explained ∼ 49% of the variance in responses strongly separates faces, animals and bodies from inanimate objects, despite vast variations in the visual appearance of images within sub-categories. The second PC, accounting for ∼ 16% variance, distinguished objects based on their shape (circular vs. rectilinear), suggesting that such shape features are important for VO. Importantly, these features do not emerge when the same analysis is applied to early visual ROIs (V1-V4).

**Response-optimized models exhibit a shape bias** A striking known discrepancy between human and machine vision is the shape vs. texture bias: human classification decisions are shape-biased whereas CNNs trained for CV tasks tend to classify images by texture [57, 58]. We study whether the shape bias of humans is reflected in response-optimized models of cortical regions that subserve visual categorization in humans. We first train a linear layer on top of the predicted responses of response-optimized models to map these responses to visual categories in the ImageNet-16 database [57] using a small subset of this database. We then assess the transfer performance of response-optimized models for classification on held-out stimuli from the ImageNet-16 dataset as well as the Silhouette dataset described in [57] (Appendix A.10). We find that the classification performance is significantly better than what would be expected from the same network architecture with a random initialization (Figure 6B), suggesting that the brain response-optimization objective yields effective representations. Further, this classification performance is

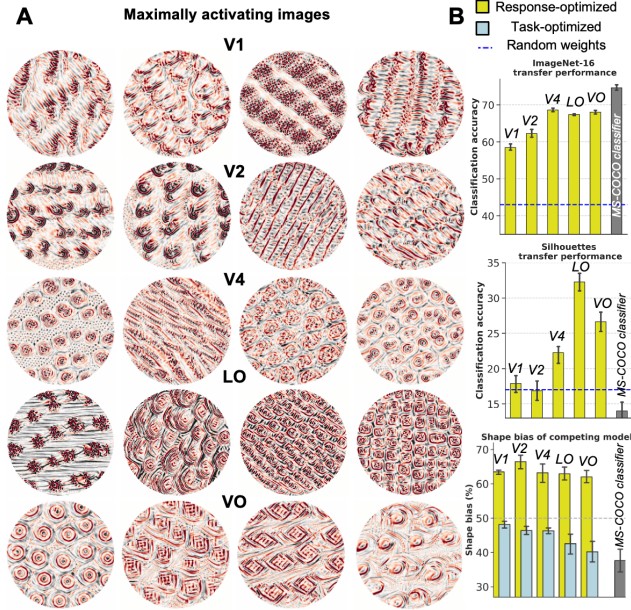

Figure 6: **A Maximally activating images** for 4 model neurons (emulating brain voxels) in each visual ROI model, generated using the image synthesis procedure. **B Shape bias analysis**. Top and middle barplots show the transfer performance of response-optimized models and MS-COCO trained object classifier on the ImageNet-16 and Silhouette datasets. Bottom plot shows the shape bias of all response-optimized and task-optimized encoding models and the MS-COCO trained multi-label object classification network.

better using models of intermediate and high-level visual areas (V4, LO and VO) than early visual areas, even approaching the transfer performance of an MS-COCO-trained object classification network

with the same architecture (Figure 6B, top, see Appendix A.10 for model details), providing further support for the separability of categorical information in high-level, but not early visual cortex [11]. The LO model has the highest classification performance on the Silhouette dataset (which solely contain shape information), consistent with and providing further evidence for previous hypotheses that LO tracks object shape information [59]. We then evaluate the classification performance of this mapped response-optimized model on the diagnostic texture-shape cue conflict dataset [57] to assess whether the model makes its categorization decisions based on shape or texture (Appendix A.10). We follow [57] and define *shape bias* as the percentage of times the model classifies an image based on its shape over texture, provided it returned either the correct shape or correct texture label for the image. A model is shape-biased if its *shape bias* $> 50\%$. We observed that *all* response-optimized models exhibit a shape bias (Figure 6B [bottom], Appendix Figure A.1), with this metric ranging from 56.7-69.4%, suggesting that representations within the ventral visual stream are likely biased towards shapes than textures. This is in stark contrast to standard CV models which show a texture bias, for e.g., the same architecture trained for multi-label object classification on MS-COCO shows a strong texture bias and performs significantly worse on the Silhouettes dataset (Figure 6B, gray bar), suggesting that the distribution of NSD stimuli (which are drawn from MS-COCO) is not sufficient to drive the shape bias and the training objective (matching to neural responses rather than object category labels) matters. Further, as shown in Figure 6B (bottom), it is not the linear mapping to neural response but the way features are optimized that explains the differing shape/texture bias of task-optimized and response-optimized encoding models. We next performed a *feature visualization* analysis to find visual patterns that maximally activate individual model neurons emulating brain voxels in response-optimized encoding models. Following [60], we start with a random noise input $x_0$ for each model neuron $i$ and iteratively update the input along the gradient $\frac{\partial a_i}{\partial x}$ to synthesize inputs that would result in higher and higher predicted activation $a_i$ for that neuron (Appendix A.9). Visual inspection shows that the features increase in complexity as we go from early to high-level visual ROI models: simple edge-like features or high-low frequency detectors are seen in V1, curves and edges are seen in V2, more complex curvatures in V4 and finally, fully-formed complex shapes like concentric hexagons are seen in LO and VO (Fig. 6A, Appendix Fig. A.7). We also validated this increased complexity using quantitative measures based on compression ratios (Appendix Table A.3).

## 4  Discussion

In this study, we leveraged the large sample size and natural variation represented in the Natural Scenes Dataset to train deep neural networks directly on the brain response. Concurrently, recent papers [31, 30] have also demonstrated that when trained with large amounts of data such as NSD, the response-optimization approach can perform competitively with state-of-the-art task-optimized models (discussed in more detail in Appendix C.1). We studied model generalization in various settings: from synthetic stimuli to novel fMRI datasets with different preprocessing. Our results recapitulated key features of early visual areas (e.g. their precise retinotopic organization) and representational properties in higher level areas. We also demonstrated that response-optimized models can capture human behavior, including not only object categorization, but also other fine-grained measures such as perceived image similarity. The response-optimization approach, in principle, allows exploring infinite paths in the model space, and when optimized properly, converging towards one that explains maximum possible variance in the data. Since any features that emerge in these optimized networks are a result of optimization to match neural targets, unconfounded by any other supervision signal, these models can provide stronger tests about the visual representations within different visual areas.

In terms of human object categorization behavior, our models lack significantly compared to ImageNet-supervised DNN models. While part of this may be due to the domain shift, the restricted transfer learning setting (a single linear layer mapping), small size of the transfer set or simply the coarse spatial resolution of fMRI, this gap is worth exploring. Further, using neural data to build better models for other communities (e.g. computer vision) is another scientifically fascinating direction and can develop reinforcing cycles. Studies have shown that explicitly encouraging neural networks to build representations similar V1 representations can improve their robustness under noise and adversarial attacks [61, 62]. In a similar vein, regularization with high-level visual cortical activity might help grant models brain-like biases (e.g. shape bias) and close the gap between human and machine vision. Alternatively, probing the connections in response-optimized models, which already exhibit a shape bias, using model pruning and attribution methods can yield a mechanistic understanding of the shape bias in neural activity and reveal principles that may enable CV models

to attain similar representations. Recent studies have inspired optimism that studying connection weights to reveal underlying computational principles may be tractable [34, 63].

## Acknowledgements

This work was funded by the following grants: NIH RF1 MH123232, NIH R01 LM012719, NIH R01 AG053949, NSF CAREER 1748377, and NSF NeuroNex Grant 1707312 (MS).

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
