# Appendix

## A Materials and Methods

### A.1 Natural Scenes Dataset

Here, we briefly summarize the data acquisition and preprocessing steps that are described in detail elsewhere [1]. Scanning was conducted at 7T using whole-brain gradient-echo EPI at 1.8-mm resolution and 1.6-s repetition time. Images were taken from the Microsoft Common Objects in Context (COCO) database [2], square cropped, and presented at a size of 8.4° x 8.4°. A special set of 1,000 images were shared across subjects; the remaining images were mutually exclusive across subjects. Images were presented for 3 s with 1-s gaps in between images. Subjects fixated centrally and performed a long-term continuous recognition task on the images. The fMRI data were pre-processed by performing one temporal interpolation (to correct for slice time differences) and one spatial interpolation (to correct for head motion). A general linear model was then used to estimate single-trial beta weights. Cortical surface reconstructions were generated using FreeSurfer, and both volume- and surface-based versions of the beta weights were created. Estimated beta weights provided in the subject-native space (func1pt8mm) were used in all of our experiments. Every stimulus considered in this study had 3 repetitions. We averaged single-trial betas after z-scoring every voxel within each scan session to create our voxel responses.

### A.2 Response-optimized Model Training Routine

Response-optimized models for each visual area were trained for a maximum of 100 epochs using Adam with a learning rate of 1e-4, a batch size of 16 and early stopping (patience = 20) based on the Pearson's correlation coefficient between the predicted and measures responses on the validation set. The target signals for each model comprised the image stimuli and voxel-level response vector for each ROI. The loss is computed as the following:

$$\mathcal{L} = \sum_{i \in \text{Batch}} \sum_{s=1}^{S} \sum_{v=1}^{n_s} 1_{i \in I_s} \left( r_{s,v}^{pred} - r_{s,v}^{meas.} \right)^2,$$

where $1_{i \in I_s}$ is the indicator variable specifying if image $i$ was shown to subject $s$, $r_{s,v}^{pred}$ and $r_{s,v}^{meas.}$ are the predicted and measured response at subject $s$, voxel $v$. This masked squared error loss allowed us to backpropagate errors through the shared convolutional backbone even if the subjects were not exposed to common stimuli. The other response-optimized baseline model (denoted as 'Response-optimized (no rotation symmetries)' in the main text) is optimized following the same training procedure as defined above.

### A.3 Details of the task-optimized DNN models evaluated

Table A.2 lists all DNN models used as baselines against response-optimized models for quantitative comparisons. All pre-trained models, except CORnet-S were downloaded from the official PyTorch Model Zoo. CORnet-S was obtained from the official github repository of the project [3]. For evaluation of all models except CORnet-S, we downsample convolutional layers by applying an avgpool layer so that the resulting feature map has dimension at most 9K. We found that further decreasing it (e.g. to 1K) only decreased the prediction accuracy on the validation set. For CORnet-S, we do not perform layer selection for each of the 5 visual regions, rather we pre-select the layer homologous to each visual region, i.e., layers V1, V2 and V4 are used for modeling responses to voxels in V1, V2 and V4 respectively. We further use the layer IT for modeling responses to LO and VO. The pre-selection of convolutional layers further allows us to employ a spatial x features factorized linear readout for the CORnet-S model to enable a fairer comparison with the proposed response-optimized encoding models. We further resize the convolutional output of the CORnet-S model to have spatial dimensions of $28 \times 28$ (same as the response-optimized model).

| Quantitative fit (response predictivity) | | | |
|---|---|---|---|
| Analysis | Probe/evaluation dataset | Metrics for evaluation | Baselines |
| NSD (i) Generalization to *novel stimuli* (in-distribution) | Held-out stimuli-response pairs from the same NSD subjects | Prediction accuracy (Pearson's R) | Category ideal observer, ImageNet-trained models (AlexNet, CORnet-S, ResNet-50, DenseNet121) and MS-COCO-trained (ResNet-50) task-optimized models) |
| (ii) Generalization to *novel subjects* | Held-out stimuli-response pairs from 4 novel NSD subjects 'NSDsynthetic' stimuli-response pairs | | |
| (iii) Generalization to *OOD stimuli* | | | |
| Generalization to novel fMRI datasets | Algonauts 2019, BOLD5000, Inanimate objects dataset | RSA (Kendall's $\tau$) | Comparison among response-optimized models |
| Characterizing neural response properties by probing response-optimized models | | | |
| Characterizing spatial tuning | Measured pRF maps from NSD Retinotopy experiment | Pearson's R (Eccentricity); Circular correlation coefficient (Polar angle) | — |
| Characterizing feature representations (i) Alignment with human perception | NSD-meadows multi-arrangement task | RSA (Kendall's $\tau$) | Comparison among response-optimized models |
| (ii) Separability of category information | THINGS, ImageNet-16, NSD | RSA (Kendall's $\tau$) | |
| (iii) Single voxel tuning (maximally activating images) | – | – | |
| (iv) shape v/s texture bias | ImageNet-16, Silhouette and Geirhos Style Transfer dataset | Tranfer performance (classification accuracy), shape bias | MS-COCO-trained multi-label object classification network, task-optimized encoding model (AlexNet) |

Table A.1: Summary of model evaluation and interpretability techniques employed in this study

| Name | Number of parameters | Layers selected for evaluation |
|---|---|---|
| **AlexNet** | 0.61M | conv1, conv2, conv3, conv4, conv5, fc6, fc7 |
| **CORnet-S** | 53.4M | V1, V2, V4, IT |
| **ResNet-50** | 25.6M | Last layer of every residual stage (res1, res2, res3, res4) and avgpool |
| **DenseNet-121** | 8M | Last layer of every dense block (denseblock1, denseblock2, denseblock3, denseblock4) |
| **ResNet-50 backbone (FasterRCNN, MS-COCO)** | 41.7M | Last layer of every residual stage (res1, res2, res3, res4) and avgpool |

Table A.2: Details of the task-optimized DNNs used as baselines

## A.4  Characterizing the spatial tuning of early visual areas: Polar angle agreement

We measured the agreement between the polar angles estimated from the learned encoding model against the polar angle measurements from the independent pRF experiment using the circular correlation coefficient. The circular correlation coefficient between measured and predicted polar angle arrays in an ROI of *n* voxels, respectively denoted by $\{a_m^1, .., a_m^n\}$ and $\{a_p^1, .., a_p^n\}$, called '*polar angle agreement*', is calculated as,

$$r = \frac{\sum_{i=1}^n \sin(a_m^i - T_m) \sin(a_p^i - T_p)}{\sqrt{\sum_{i=1}^n \sin^2(a_m^i - T_m) \sum_{i=1}^n \sin^2(a_p^i - T_p)}},$$

where $T_m$ and $T_p$ are the circular mean angles of the measured and predicted polar angle vectors, respectively,

$$T_m = \left(\frac{1}{n} \sum_{i=1}^n \sin a_m^i, \frac{1}{n} \sum_{i=1}^n \cos a_m^i\right), T_p = \left(\frac{1}{n} \sum_{i=1}^n \sin a_p^i, \frac{1}{n} \sum_{i=1}^n \cos a_p^i\right)$$

## A.5  Noise ceiling estimation

Noise ceiling for every voxel represents the performance of the "true" model underlying the generation of the responses (the best achievable accuracy) given the noise in the fMRI measurements. They were computed using the standard procedure followed in [1] by considering the variability in voxel responses across repeat scans. The dataset contains 3 different responses to each stimulus image for every voxel. In the estimation framework, the variance of the responses, $\sigma_{\text{response}}^2$, are split into two components, the measurement noise $\sigma_{\text{noise}}^2$ and the variability between images of the noise free responses $\sigma_{\text{signal}}^2$.

$$\hat{\sigma}_{\text{response}}^2 = \hat{\sigma}_{\text{signal}}^2 + \hat{\sigma}_{\text{noise}}^2$$

An estimate of the variability of the noise is given as $\hat{\sigma}_{\text{noise}}^2 = \frac{1}{n} \sum_{i=1}^n \text{Var}(\beta_i)$, where i denotes the image (among $n$ images) and $\text{Var}(\beta_i)$ denotes the variance of the response across repetitions of the same image. An estimate of the variability of the noise free signal is then given as,

$$\hat{\sigma}_{\text{signal}}^2 = \hat{\sigma}_{\text{response}}^2 - \hat{\sigma}_{\text{noise}}^2$$

Since the measured responses were z-scored, $\hat{\sigma}_{\text{response}}^2 = 1$ and $\hat{\sigma}_{\text{signal}}^2 = 1 - \hat{\sigma}_{\text{noise}}^2$. The noise ceiling (n.c.) expressed in correlation units is thus given as $n.c. = \sqrt{\frac{\hat{\sigma}_{\text{signal}}^2}{\hat{\sigma}_{\text{signal}}^2 + \hat{\sigma}_{\text{noise}}^2}}$. The models were evaluated in terms of their ability to explain the average response across 3 trials (i.e., repetitions) of the stimulus. To account for this trial averaging, the noise ceiling is expressed as $n.c. = \sqrt{\frac{\hat{\sigma}_{\text{signal}}^2}{\hat{\sigma}_{\text{signal}}^2 + \hat{\sigma}_{\text{noise}}^2/3}}$. We computed noise ceiling using this formulation for every voxel in each subject and expressed the noise-normalized prediction accuracy (R) as a percentage of this noise ceiling.

## A.6  Principal components analysis for characterizing features in high-level visual area VO

We implemented a principal components analysis on the predicted activity patterns in the anterior ventral stream ROI VO to all 26,107 images from the THINGS dataset [4]. This also enables us to find a compressed representation of thousands of voxels in VO (across 4 subjects). We found that the first 2 principal components explained a large proportion of the variance in voxel responses (main text Figure 5D). We examined the 2 discovered components by ordering exemplars along each component dimension for intuitive exposition. Specifically, we extracted the top and bottom 100 images for each PC dimension. We extracted the top 5 concepts that had the highest frequency in each of the top and bottom image sets (the THINGS dataset comprises a total of 1,854 diverse concepts) and visualized the two images within each of these concept categories that elicited the highest/lowest response along the PCs. These images are shown in the main text Figure 5D [left]. We also plotted all images in the 2-dimensional space spanned by the PCs and colored them based on the animate/inanimate labels provided along with the THINGS dataset (as part of the Top-Down WordNet Category ).

## A.7 Generalization to held-out NSD subjects

For each of the 4 held-out NSD subjects, the dataset comprised brain responses to 5,445 stimuli with 3 repetitions (remaining stimuli had either 1 or 2 repetitions and were discarded from analysis). To assess generalization of response-optimized models in terms of predicting responses on held-out NSD subjects, we fixed the weights of the shared convolutional backbone and used data from new NSD participants to train their linear readouts. We varied the size of the training set for learning readout weights from a mere 100 stimulus-response pairs from every new participant to a large set of 4,500 pairs. The readouts had the same architecture as the original models, i.e., they were factorized into spatial and feature dimensions. The readouts were trained independently for each subject, each training set size and each visual ROI for 100 epochs with an early stopping criterion (patience of 20). For task-optimized and category ideal observer models, we fit linear regression models (as described above for the original 4 subjects) using these restricted sample sizes. The regularization parameter for both semantic and task-optimized models was optimized independently for each training set size, each subject and for voxels in each visual area by testing among 8 log-spaced values in [1e-4, 1e4].

## A.8 Category ideal observer model

We fit a category ideal observer model using category annotations for NSD images. The input to the categorical model is an 80-D binary vector corresponding to the 80 object categories annotated in the MS-COCO database, where each element indicates whether the corresponding category was present in the image or absent. The weights corresponding to different categories for every voxel are optimized by fitting a $l_2$ regularized linear regression model. The regularization parameter for this model was optimized independently for each subject and for voxels in each visual area by testing among 8 log-spaced values in [1e-4, 1e4].

## A.9 Maximally activating images

We performed k-means clustering (k=4) on model-predicted voxel-level responses of each visual ROI to locate 'representative' voxels (cluster medians) for visualization, rather than randomly selecting these voxels. Following [5], we start with a random noise input $x_0$ for each model neuron $i$ and iteratively update the input along the gradient $\frac{\partial a_i}{\partial x}$ to synthesize inputs that would result in higher and higher predicted activation $a_i$ for that neuron. For visualization purposes, since we were interested in *featural* tuning, we discarded the spatial mask in the readout of every voxel and used the learned feature tuning of every voxel to create an additional 1x1 convolutional layer, so that every voxel is represented by an independent unit in this convolutional layer and synthesized inputs to activate individual units in this convolutional *voxel* layer instead. Most visualization techniques further employ an *image prior* in the form of a regularizer to encourage stable results [6, 7]. This visualization technique is commonly employed in neural network interpretability research to find the features that drive model neurons. Formally, the goal of finding the maximally activating image $x^*$ is then expressed as the following optimization problem.

$$x^* = \underset{x \in \mathbb{R}^{H \times W \times C}}{} A_{ij}(\theta, x) + \mathcal{R}(x)$$

where $A_{(i,j)}(\theta, x)$ denotes the activation of unit $i$ from layer $j$ in the neural network to input $x$ (H: Height, W: Width, C: Channels), and $\theta$ denotes the parameters of the network. The latter are fixed during the above optimization procedure. $\mathcal{R}(x)$ denotes the regulariser. In order to generate maximally activating image for the $j$th voxel, we set $i$ to the output voxel layer and $j$ to be the index of the model neuron in the output layer that emulates voxel $j$. We find a local solution for the above optimization problem by performing gradient ascent in the input space and updating $x$ iteratively in the direction of the gradient of $A_{ij}(\theta, x) + \mathcal{R}(x)$. We employed a very weak form of regularization, where we stochastically jitter (up to 3 px)) the image before each optimization step to avoid high frequency noise. We also blurred the image after every gradient ascent step using a Gaussian filter to avoid high-frequency effects. The images are optimized starting from random noise with Adam optimizer for 1000 steps using a step size of 1.

## A.10 Shape bias analysis

**Training baseline object classification model on MS-COCO**  We train a baseline DNN for multi-label object classification on the entire MS-COCO dataset. The dataset comprises 82,081 training and

40,137 validation images, with most images containing multiple objects at once. Since NSD images were drawn from the MS-COCO dataset, this serves as a useful control for assessing features that may arise simply due to the dataset distribution. The model employs the same backbone architecture as the proposed rotation-equivariant response-optimized model comprising 4 convolutional blocks with E(2)-steerable convolutions. The convolutional output is downsampled to a spatial resolution of $4\times4$ and is mapped on to an 80-D output using a linear layer. The model is optimized for multi-label classification using a Binary Cross-Entropy loss with an Adam optimizer (learning rate of 0.001). The model is trained for 30 epochs and achieves a mean Average Precision (mAP) measure of 66.7% on the validation set, which is within the range of competitive CNN models, although on the lower end of that range [8].

**Mapping predicted voxel-wise responses to ImageNet-16 category labels and measuring transfer performance**   We use the proposed response-optimized models to extract the *predicted* responses of voxels in every region to stimuli from a small subset of the large-scale ImageNet-16 dataset. This subset comprised 500 images from each of the 16 categories, resulting in a total of 8,000 images (same samples were used in the RDM analyses shown in the main text Figure 5B). The predicted voxel-wise responses for every visual area constitute the *representational space* of that ROI. We randomly split the 8,000 images into training (70%), validation (5%) and test sets (25%) and fitted $l_2$ regularized linear classification models (known as ridge classifiers) on top of this ROI-specific representational space to predict the category label of test images from the ImageNet-16 subset. The predicted labels were compared against the ground truth labels to compute the ImageNet-16 transfer performance. Error bars are computed over 5 random splits. This approach is akin to supervised linear probes [9], where linear classifiers are trained independently of the models to probe their neural representations. The classification accuracy provides a quantitative measure of the ability of each ROI-specific neural representational space to support object classification. As demonstrated in the main text, this classification capacity increases along the ventral visual hierarchy 4.

We also used the response-optimized models to extract the model-*predicted* responses of voxels in every region to 160 images (10 per category) from the Silhouettes dataset [10]. The predicted responses were mapped onto the 16 category labels using the linear classifier weights estimated with the ImageNet-16 dataset (as described above) without any additional fitting. This also helps us assess how well the prediction function generalizes to out-of-domain stimuli, such as image silhouettes. The classification performance on this dataset is termed 'Silhouettes transfer performance'.

**Shape bias evaluation**   (a) Response-optimized models: Finally, we use the response-optimized models to extract the model-*predicted* responses of voxels in every region to 1200 images from the Geirhos cue-conflict dataset [10]. We map these predicted responses onto the 16-class-ImageNet categories using the linear classifier weights estimated with the ImageNet-16 subset (as described above). Each image from this dataset has a texture and a shape label. The shape bias of each response-optimized model is then computed as the percentage of times it classifies images from the cue-conflict dataset according to shape, provided it classified either shape or texture correctly.

(b) Task-optimized models: We can also compute the shape bias of task-optimized encoding models (here, we assess the AlexNet model) by extracting their predicted voxel-wise responses to the images from the Geirhos cue-conflict dataset and mapping these responses to the 16-class-ImageNet categories. For the mapping, we follow the same procedure as employed in the case of response-optimized models, wherein a linear classifier is first trained to map the predicted responses to 16 classes using the 8,000 images from the ImageNet-16 subset. The transfer

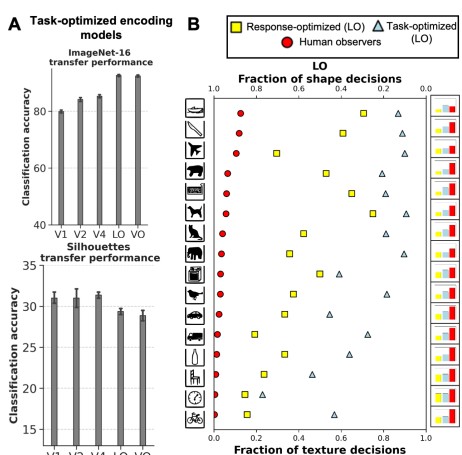

Figure A.1: **A.** Transfer performance of different task-optimized encoding models on ImageNet-16 dataset. **B.** Fraction of shape vs. texture decisions for stimuli with cue conflict. Bar plots on the right display the proportion of correct decisions (either shape or texture recognized correctly) as a fraction of all trials for human observers as well as response-optimized and task-optimized models of LO.

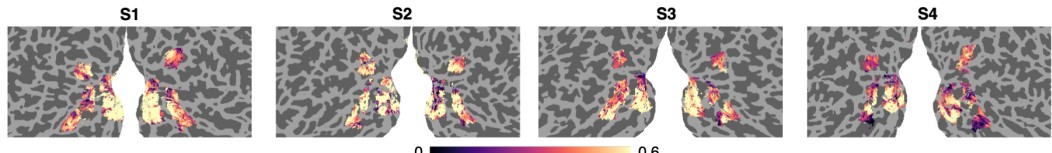

Figure A.2: *Voxel-wise prediction accuracy (R)* of the proposed response-optimized model across all voxels in all the 5 visual ROIs depicted on cortical flatmaps for each subject. High prediction accuracy ($> 0.6$) is obtained across large swathes of the cortex, well beyond early visual cortex (V1-V4) into LO and VO as well.

performance in this case is much higher than the
transfer performance of response-optimized models,
as shown in Supplementary Figure A.1. This gap is
perhaps unsurprising given that the task-optimized
encoding models inherit feature representations *optimized* for ImageNet classification. The shape
bias is evaluated using the same approach as response-optimized models. As shown in the main
text Figure 4B[bottom], these models exhibit a strong texture bias. This is also not surprising given
that the task-optimized encoding model is more likely to reflect the biases of supervised ImageNet
training [10].

## B Supplemental Results

### B.1 Voxel-wise prediction accuracy on cortical surface

Supplementary Figure A.2 depicts the raw voxel-wise prediction accuracy of the proposed response-optimized models on the cortical surface for each of the 4 NSD participants analyzed in the main
study.

### B.2 Generalization to novel datasets

We also compared the representational similarity structure captured by response-optimized models of
different ROIs to novel stimuli from other well-known fMRI datasets besides NSD. Here, we wanted
to assess whether the models indeed captured ROI-specific features that could generalize in explaining
representational geometries of similar visual ROIs in different datasets. While the ROI definitions
and nomenclature can vary substantially across these datasets because of the different protocols
involved in their localization, we expected the response-optimized models to capture representational
geometry of voxel responses in ROIs that lie in a roughly similar anatomical location as any of the
5 ROIs considered in this study. We first performed this generalization analysis using the classical
RSA framework (without any fitting) on the following two widely-used fMRI datasets: (1) **Cichy
et al. dataset [11]** comprises fMRI recordings while 15 human participants observed 92 images of
natural objects (this dataset was also used in the Algonauts 2019 Challenge Training Set): Here, the
subject-averaged RDMs of early ('EVC') and high-level visual ROIs ('IT') were already computed
and publicly distributed by the authors. Further details about the localization of these 2 ROIs are
provided in [11]. (2) **BOLD5000 V2 (GLM-denoised) dataset [12]** comprises fMRI recordings
from 4 subjects (CSI1-4), while they each viewed ~5,254 natural scene images. Subject CSI4 was
discarded from this analyses since this subject completed much fewer scan sessions. We restrict our
analyses to the 1,000 images within the BOLD5000 dataset that are shared with the NSD dataset.
This enables us to compute the match not just against model-predicted RDMs but also against RDMs
derived from *measured* responses in NSD subjects. Importantly, these 1,000 images are all part of the
test set and were not used in training any of the response-optimized models. We focus on two visual
ROIs in this dataset, namely 'EarlyVis' and 'LOC', which had (at least) a partial overlap with some
of the 5 visual ROIs we modeled in our study.

For stimuli from each of the above datasets, we first extracted model-predicted responses from
the response-optimized models of all 5 visual ROIs: [V1, V2, V4, LO, VO]. We then computed
model RDMs separately for each ROI by computing the pairwise correlation distance between the
model-predicted responses of all images in each dataset. We then computed the RDM similarity
between model-predicted RDMs of each ROI against the observed-response RDMs (computed using

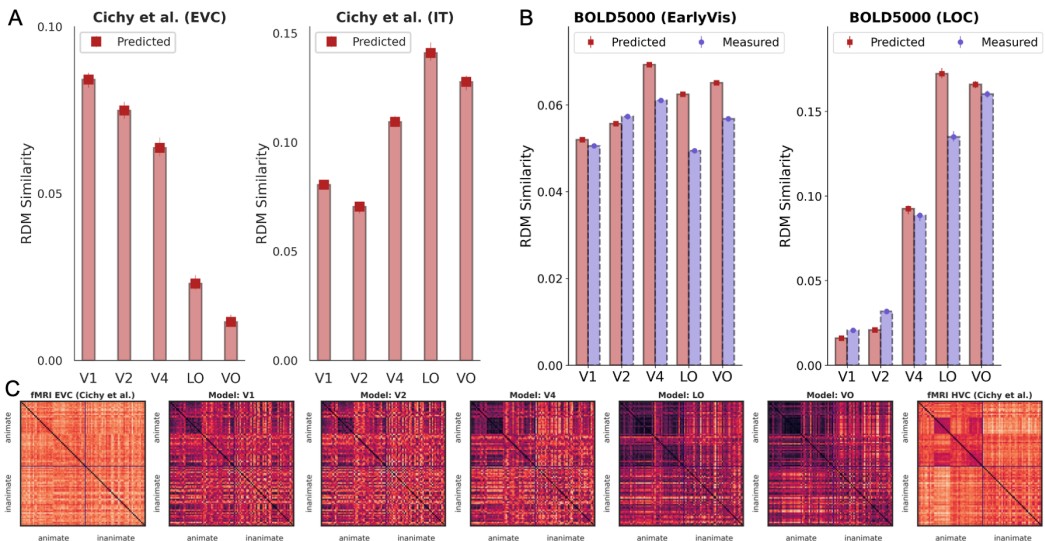

Figure A.3: *Generalization to the Cichy et al. and BOLD5000 datasets* assessed with classical RSA. A and B depict the RDM similarity between model-predicted RDMs (5 models corresponding to the 5 visual ROIs considered in this study) and fMRI RDMs from the Cichy et al. and BOLD5000 datasets, respectively. C depicts the model-predicted RDMs from each ROI model (V1-VO, center) as well as the fMRI RDMs of 'EVC' (left) and 'IT' ROIs (right)

the respective fMRI datasets). RDM similarities were computed using the Kendall's $\tau$ coefficient. As shown in Supplementary Figure A.3, response-optimized model RDMs captured the representational geometry of respective ROIs in these novel fMRI datasets. For e.g., early visual ROI model RDMs (V1-V4) better match fMRI RDMs of early visual areas ('EVC' in Cichy et al. and 'EarlyVis' in BOLD5000) and higher-order ROI model RDMs (LO and VO) better match fMRI RDMs of higher-order visual areas ('IT' in Cichy et al. and 'LOC' in BOLD5000). This can also been qualitatively in the model-predicted and fMRI RDMs for the Cichy et al. dataset (Supplementary Figure A.3C). Importantly, we also had access to the measured responses in NSD participants to the restricted BOLD5000 image set since the same stimuli were also shown to NSD participants. This enabled us to compare subject-averaged measured RDMs (from observed responses of NSD participants) against BOLD5000 observed RDMs. Intriguingly, this agreement against observed BOLD5000 RDMs was greater with *predicted* than *measured* NSD responses, likely highlighting the denoising quality of encoding models (Figure A.3B). Further, the model-predicted RDM similarity follows the same pattern as NSD-measured RDM similarity across ROIs, suggesting that the models indeed capture features idiosyncratic to each visual ROI.

We also assesses model generalization to a challenging new fMRI dataset, namely the Inanimate Objects dataset from Konkle et al. ( [13]). As shown in [13], current SOTA supervised and unsupervised models trained on ImageNet struggle to predict voxel responses to stimuli from this dataset. This dataset comprises fMRI responses from 10 participants to 72 everyday objects. We focused on responses within two high-level visual ROIs in this dataset, namely posterior occipito-temporal cortex (pOTC), and the anterior occipito-temporal cortex. We extracted model-predicted responses from the response-optimized models of all 5 visual ROIs: [V1, V2, V4, LO, VO] to each of the 72 images in this dataset to construct the model representational space. Model-brain match in this dataset was then measured using a voxelwise encoding RSA (veRSA) procedure, following the same protocol described in [13] for consistency. Performance of competing models, namely (i) a Supervised ImageNet-trained model and (ii) a self-supervised ImageNet-trained model, called instance-prototype contrastive learning (IPCL) with an AlexNet architecture was already reported in the main paper describing this dataset [13] and is included in Supplementary Figure A.4 for comparison. As shown in Supplementary Figure A.4, response-optimized models of high-level visual ROIs (and not early visual ROIs) perform on par with these SOTA models in predicting neural responses in both the high-level visual ROIs (pOTC and aOTC), serving as another useful model class for studying high-level visual cortical representations.

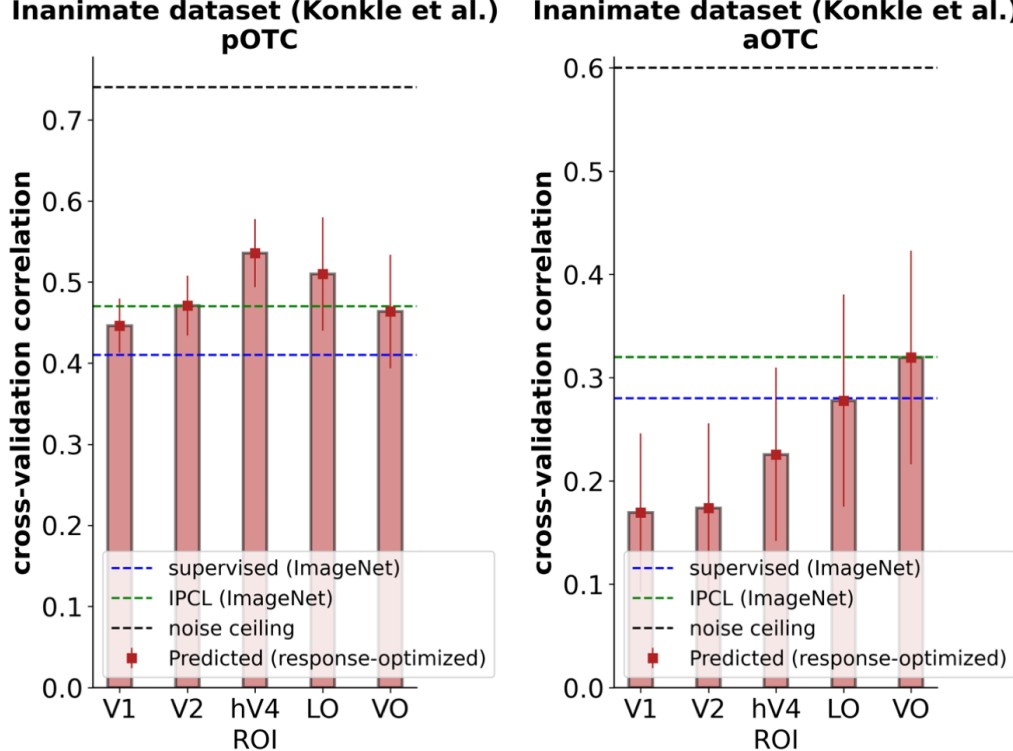

Figure A.4: *Generalization to the Konkle et al. Inanimate Objects dataset* assessed with voxel-wise encoding RSA as described in [13]. Performance numbers for competing models (Supervised (ImageNet), IPCL (ImageNet)) as well as the noise ceiling are taken directly from the values reported in the main paper (details provided in [13].)

### B.3 Models retain signatures of voxel-level idiosyncracies

We correlated the predicted response of every voxel (from the proposed response-optimized model) against the measured response of every other voxel (across all 4 subjects in the main study) in each ROI to obtain a *voxel identifiability matrix* per ROI. Visualizing this matrix helps us assess whether the models indeed captured meaningful voxel-level idiosyncracies. We note that all response-optimized models retain signatures of individual voxel-level idiosyncracies as indicated by the prominent diagonal nature of the *voxel identifiability matrices* (Supplementary Figure A.5); this illustrates that the predicted response for a voxel best matches the measured response for the *same* voxel in the *same* subject. This enables us to perform population-level analysis with these predictive models.

### B.4 Variability in voxel response predictivity is driven by the noise ceiling

We visualized the raw predictive accuracy (R) for every voxel, as achieved using the proposed response-optimized model, against that voxel's corresponding noise ceiling to see if there was a systematic trend. We observed that a large proportion of the variance in predictive accuracy across voxels is driven by their noise ceiling, as shown in Supplementary Figure A.6.

### B.5 Quantifying the complexity of synthesized images

We measured the complexity of synthesized images optimized to maximally activate individual voxels (as shown in Figure 4A) using a measure based on compression ratios, that was previously adopted to quantify complexity of synthesized images [14]. Briefly, this metric, based on the discrete cosine transform, estimates the minimum number of coefficients (corresponding to different spatial frequencies) needed to accurately reconstruct an image, which in turn indicates the 'compressibility' (or inversely, complexity) of the image. We quantified the complexity of all images in Figure 4A as

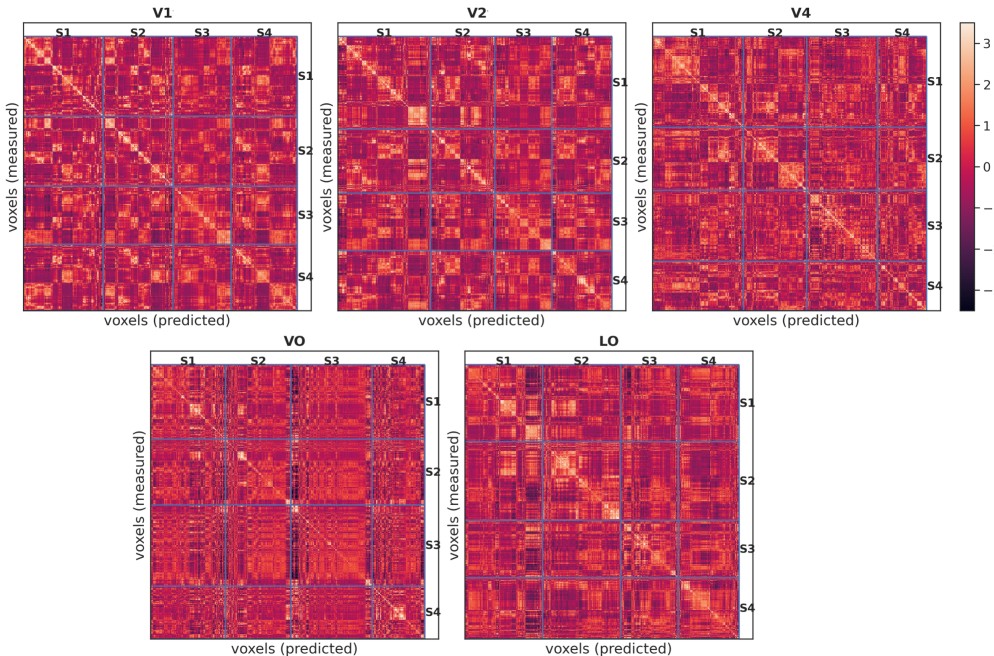

Figure A.5: *Voxel identifiability matrix* for every ROI is computed by correlating the predicted response for every voxel against the measured response of every other voxel in that ROI. To account for higher variability in measured versus predicted responses, we normalize the rows and columns of this correlation matrix.

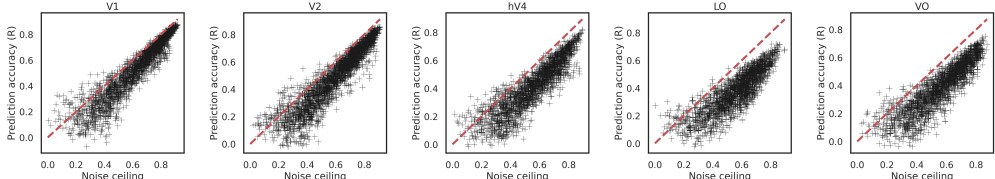

Figure A.6: Each scatter plot depicts the raw prediction accuracy (R) for all voxels (obtained using the proposed response-optimized model) belonging to one of the five ROIs against their corresponding noise ceiling.

the inverse of their respective compression ratios and found that the complexity does indeed increase along the ventral visual hierarchy A.3.

We also extracted the top 3 natural images among the THINGS database (∼27,000 images) that produce the highest predicted response for each V1 or VO voxel (the two ends of the hierarchy) visualized in Figure 4A. The natural images contain similar featural configurations as their corresponding synthesized counterparts A.7: the natural images for V1 voxels contain objects at specific orientations or high frequency spatial patterns reminiscent of the orientation preferences visible in the synthesizes images, whereas the images for VO contain complex shapes like concentric circles, rectangles and hexagons, again in agreement with their respective synthesized images.

### B.6    Human behavioral data from 'nsdmeadows'

We also employ the T-distributed Stochastic Neighbor Embedding (t-SNE) algorithm to visualize human behavioral data from the 'nsdmeadows' experiment where NSD participants performed a Multiple arrangement Task [1]. Supplementary Figure A.9 depicts the t-SNE visualization of the mean pairwise dissimilarity matrix across all 8 NSD subjects. The pairwise dissimilarity matrix (RDM) for

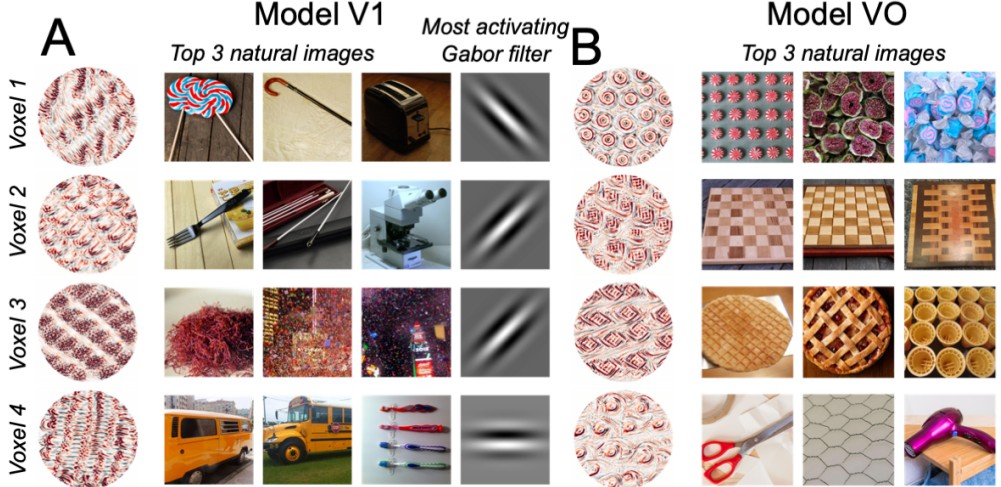

Figure A.7: **A** and **B** show the synthesized and natural images (among the THINGS database) that most highly activate each of the 4 individual voxels visualized in Figure 4A, belonging to V1 and VO, respectively. The natural images are reminiscent of the patterns in their corresponding synthesized counterparts. For V1 voxels, we also visualize the Gabor filter (among a filter bank of 4 sine and cosine gabors) that produces the highest model-predicted response for each voxel.

| Model ROI | Complexity |
|-----------|------------|
| V1 | 0.439+/-0.041 |
| V2 | 0.433+/-0.012 |
| V4 | 0.449+/-0.034 |
| LO | 0.536+/-0.079 |
| VO | 0.637+/-0.012 |

Table A.3: Complexity of the synthesized images for each visual region.

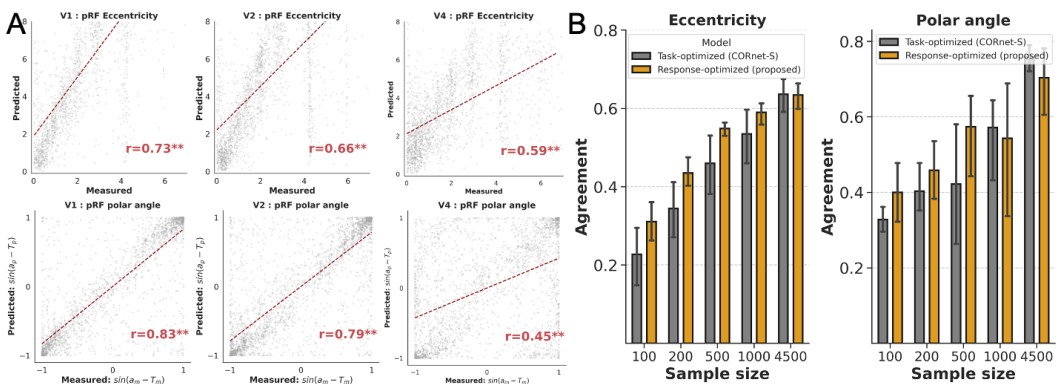

Figure A.8: **pRF estimation with task-optimized CORnet-S encoding models**. **A** Scatter plots showing predicted and localizer-estimated retinotopic parameters for all voxels in all 4 subjects. Inset correlations are computed using voxels from all 4 subjects. **C** Agreement between estimated and measured retinotopic maps as a function of training examples (stimulus-response pairs) from novel subjects used to train their linear readout. Error bars depict the 95% CI around estimated mean.

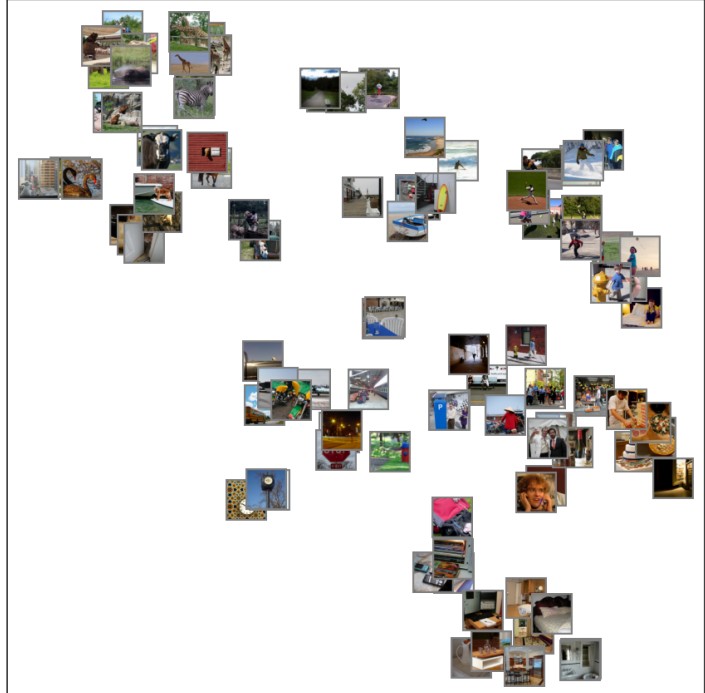

Figure A.9: **Human behavioral data** T-SNE plot depicting human similarity assessments for the 100 images used in the 'nsdmeadows' experiment.

each participant was released with NSD. We see that even qualitatively, pairwise similarities from human behavioral data appear remarkably similar to pairwise similarities estimated from model-predicted VO responses (shown in the main text Figure 5A); in both cases, faces and scenes appear clustered together. This is in contrast to the pairwise similarities estimated from model-predicted V1 responses, as shown below in Supplementary Figure A.10.

## C Supplemental Discussion

### C.1 Relationship to concurrent work

Concurrent with the present work, recent papers [15, 16] have demonstrated that when trained with large amounts of data such as NSD, the response-optimization approach can perform competitively with state-of-the-art task-optimized models. Khosla et al. [16] adopt similar neural network architectures and compare asymptotic performance and sample complexity in models of specialized category-selective regions of the brain, such as FFA and EBA, and do not study the ventral visual cortical hierarchy (the ROIs investigated in this study). St. Yves et al. [15] study the early to intermediate visual cortex (V1-V4), specifically the role played by the brain hierarchy in training brain-optimized models that are not hierarchical by design, presenting alternate evidence against the hierarchical nature of neural representations. Our work differs from these studies in several respects: (i) We develop models for regions along the *entire* ventral visual hierarchy, including higher-order ROIs like LO and VO (the human analogues of IT). (ii) We study the generalization of response-optimized models to datasets beyond NSD and compare against a richer battery of task-optimized models. (iii) We demonstrate the denoising capability of models, showing that the predicted activity not only serves as a useful surrogate for the measured activity, but also yields a better match to neural data in other datasets and human behavior. (iv) While St. Yves et al. [15] also demonstrate the ability of response-optimized models to recover the retinotopic organization in early visual cortex, they do so at the qualitative level of capturing size-eccentricity relationships. In the present study, we perform rigorous quantitative comparisons and sample complexity analysis on the ability of response-optimized models to capture the precise spatial receptive field tuning of individual voxels. (v) St. Yves et al.[15] focus on network architectures and ask if hierarchy, at the level of

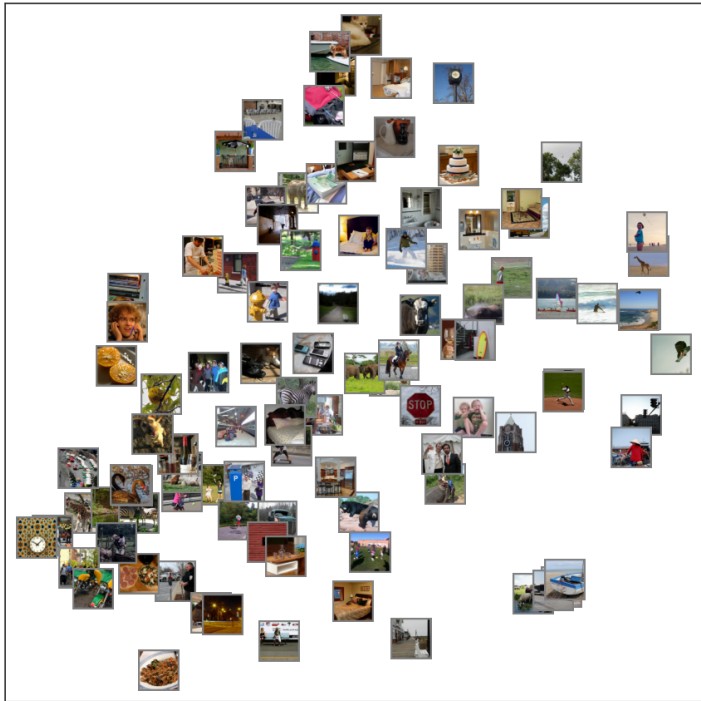

Figure A.10: **V1** T-SNE plot depicting the similarity structure of model-predicted V1 responses for the 100 images used in the 'nsd-meadows' experiment.

model architecture, is needed to explain brain activity. Here, on the other hand, we ask: what *features* emerge spontaneously in networks optimized to explain brain activity and what kinds of biases do these networks exhibit (shape v/s texture)? We demonstrate increasing separability of categorical information and alignment with human perception in the optimized *models* of ventral visual visual stream regions, providing a model abstraction of this key neural phenomenon. In the future, we hope these can be richly interrogated to generate mechanistic hypotheses. The strong shape bias in response-optimized, but not task-optimized models, further has interesting implications for computer vision models. Altogether, these studies and our work invite a shift from the task-optimized modeling framework, providing an alternate modeling strategy to understand neural representations.