# OpenReview forum: "Characterizing the Ventral Visual Stream with Response-Optimized Neural Encoding Models"
_NeurIPS.cc/2022/Conference — NeurIPS 2022 Accept_

### Official Review · Reviewer_xZPs · 2022-07-06

**Rating:** 7
**Confidence:** 3
**Soundness:** 3 good
**Presentation:** 3 good
**Contribution:** 2 fair

**Summary:**

Here, the authors leverage the Natural Scenes Dataset to train up a “hypothesis-agnostic” encoding model. In contrast to recent task-optimized models, they train a series of CNNs (one per visual area) and linear readouts (one per voxel) to predict neural responses to a large set of images. Then, generally speaking, the authors test two things about these models: (1) whether they generalize in a non-trivial way to other subjects/datasets and (2) whether the model learns interesting features in the data that may not be present in task-optimized models.

**Questions:**

1. Why was only AlexNet used as a non-trivial baseline model, since there are other CNNs already known to be better fits to neural data?
1. Why were pRFs not estimated for the task-optimized model?
1. Is it wrong to expect the task-optimized model to look very similar to what is shown in Figure 5? Should this be unique to the proposed encoding model?
1. Hopefully I didn't miss this, but did the authors measure shape bias in the neural data used to fit their encoding model? Presumably this bias is simply inherited by the encoding model?


**Limitations:**

One major limitation that I found missing in the discussion is that many of the important features found in their encoding model – e.g. correspondence to perceived image similarity – are also found in the neural data, which the models are fit to. The discussion would benefit from clarifying what unique contribution their encoding model brings to the table.

**Strengths And Weaknesses:**

Strengths

1)	The writing is clear and concise in most places.

2)	The approach to the problem – using an encoding model with a shared convolutional core – is relatively novel.

3)	Some of the results impressive, especially when the authors focus on generalization and sample efficiency.

Weaknesses

1)	It is difficult to contextualize the encoding model, especially the results of Figure 1, without a more systematic set of baseline models. While it is not necessary to comprehensively survey the field a la CORnet (Kubulius et al. 2019), it is essential to know if the gains the authors’ see are unique to their proposed encoding model. Here, I would include top performing models for ImageNet (e.g. DenseNet and ResNet) along with the older AlexNet that they have already included.

2)	As mentioned above, the authors argue that their encoding model (1) generalizes well and (2) learns non-trivial features and response properties. They present compelling evidence for (1), but (2) is less clear. For example, Figure 4 presents measured/predicted retinotopic parameters, but without the task-optimized network for comparison. Are these features unique to their encoding model? Even if the encoding model is a better match, it is explicitly optimized to better match neural data, so it hard to know whether this is truly surprising or not.

    1.	Same point holds for Figure 5 – I would expect this result in the task-optimized model as well.

    1.	More broadly, I think the point about generalization is interesting – maybe their encoding model does learn a set of features that are generically useful in a visual system. However, the fact that the model learns certain features is potentially circular since the model is optimized to mirror neural responses. In other words, it would be more surprising for a task-optimized network to have neural-like responses, while here the model is explicitly optimized to have neural-like responses.

3)	Related to the previous point, I’m a little confused about the shape bias analysis. Presumably the neural responses that the model is fit to have a shape bias – though I could not find this analysis – in which case the model should inherit a shape bias via the fitting procedure. Perhaps I am misunderstanding this, but it seems that the results in Figure 6 are potentially circular.


Minor

1)	Why is the prediction performance pooled across visual areas higher than the prediction performance for each visual area on its own?

2)	Include units in all axes for convenience (e.g. RDM similarity is Kendall’s Tau, I would include in the y axis where relevant, e.g. Supp. Fig. 3A).

3)	Should the task-optimized network  also be in Figure 6B middle – Silhouettes transfer performance?

---

> ### Author Response · Authors · 2022-08-02
> **Response to Reviewer xZPs [Part 1]**
>
> We thank the reviewer for their careful reading of our manuscript and actionable suggestions. We address their concerns below:
>
> ### Baselines for neural predictivity
> We have now included several new competitive baselines in our quantitative results for prediction accuracy to enable a better comparison with the state-of-the-art in the field. These include more recent SOTA task-optimized models, including model architectures like ResNet-50, CORnet-S and DenseNet optimized for image recognition on the ImageNet database. Since these models can, in principle, demonstrate lower prediction performance due to the domain shift between ImageNet and NSD stimuli, we also employed a ResNet-50 architecture pre-trained for object detection on the MS-COCO dataset as an additional baseline (as NSD stimuli were drawn from the MS-COCO dataset). Details of these models have been listed in Appendix Table A.2. The results confirm that the proposed response-optimized model performs competitively with these strong baselines. On the NSDsynthetic dataset, we observe that the ResNet-50 task-optimized model performs better than the response-optimized model in region VO; we have toned down our claims accordingly.
>
> ### pRFs for the task-optimized model
> We note that in the original setup, task-optimized models were trained with a full readout that was not factorized into spatial x feature dimensions. Thus, we couldn’t characterize the spatial tuning of voxels with these models directly. We have now included a CORnet-S baseline that uses the same (spatial x features) factorized readout architecture as our proposed model. We then analyzed the learned spatial readout weights of this model and quantified their agreement with localizer-estimated pRF parameters (following a similar approach as the response-optimized model). As shown in Appendix Figure A.8, while task-optimized models show a similar or slightly better match to pRF parameters on the original set of NSD subjects, there is a considerable effect of sample size (training examples) on this agreement for novel subjects; i.e., they require more training examples than the response-optimized model in order to accurately characterize retinotopic parameters in novel subjects.
>
> ### Shape bias analysis on neural data directly
> We note that it is not possible to perform the shape bias analysis on the neural data directly since it relies on independent stimuli with texture-shape cue conflict (Geirhos Style Transfer dataset), for which corresponding fMRI measurements were not available. We agree that it is likely, even suggested by our results, that shape sensitivity and shape bias is strongly represented in neural responses themselves (over texture sensitivity). This allows the model to latch on to these features and subsequent probe experiments to reveal this selectivity. This further highlights another utility of models in that they allow us to distill meaningful attributes of neural responses, and allow us to run in-silico experiments on probe stimulus sets. For instance, they enabled us to assess category separability and shape-texture bias in large-scale datasets (ImageNet, GST) for which the neural response measurements would be prohibitively expensive. They also allow us to test generalization on very different stimulus sets; this is especially relevant since some metrics like RSA are extremely sensitive to the stimuli and testing on multiple datasets can help overcome stimulus set biases and lead to more robust conclusions. We note that these results aren’t circular since these inferences are drawn on independent stimuli.
> We further note that the preferential bias for shape v/s textures is an important point of divergence between SOTA task-optimized models and human behavior. Our results suggest that the shape bias is likely reflected in the neural population responses and studying this neural basis may pave the way for models that are better aligned to human behavior.

---

> > ### Author Response · Authors · 2022-08-02
> > **Response to Reviewer xZPs [Part 2]**
> >
> > ### Limited surprise of the findings
> > We agree with the reviewer that the ability of task-optimized CV models to exhibit representations similar to the brain is a striking and powerful result. We very much appreciate the utility of task-optimized models in furthering our understanding of the visual system. But there are also significant inconsistencies between these models and human behavior and gaps remain in their respective robustness to distortions, image-level consistency, texture versus shape biases etc (Geirhos et al., 2021). In this study, we were motivated by these limitations to develop alternate models for the visual regions and study their emergent representations. We believe that the ability of response-optimized models to exhibit interesting behavioral phenomena (separable category information, increased model-human alignment, shape bias) provides a complementary perspective in that it offers a hypothesis-neutral way to study their representations and can be applied to other brain areas for which we do not know the right *task* yet. The main advantage of response-optimized networks is their flexibility. Oftentimes, we don't know what the space of likely functions a neuron might perform is. Thus, pinning down a couple hypotheses (in the form of task optimization) and devising experiments to separate them can often be sub-optimal. The stronger shape bias observed in these models versus the texture bias of modern CV task-optimized models further reveals the neural basis for the human behavioral phenomenon of shape bias. These results could be potentially interesting for the CV community as well.
> >
> > ### Interpretation of response-optimized and task-optimized models
> > In principle, we can perform the same visualization experiments (Figure 5) on task-optimized models too. But, there is an important difference. Any features that emerge in the response-optimized model are the sole result of matching to a target neural response, thus we can more confidently attribute the optimized images/feature maps to the neural population. Task-optimized models, on the other hand, inherit the biases of pre-training on ImageNet. In the task-optimized setup, features are captured by an independent network and are pre-determined, leaving the layer selection and linear mapping as the only control knobs. Having a fixed representation space means that model interpretation  can be confounded by the biases imposed by the pre-training task and dataset. We were also motivated by the following shortcomings of task-optimized encoding models with regards to model interpretation.
> > - Which layer best matches a given brain region, has been found to depend on the resolution of the image fed to DNNs (Cadena et al., 2019). This suggests that deeper layers can exhibit a stronger match to deeper regions, simply because of their increased receptive field sizes and increased *feature* complexity can play a secondary or a minimal role in the correspondence. Thus, interpreting these models will likely not be as informative as interpreting a response-optimized model, since any set of features that emerge in the response-optimized networks are optimized to explain representations in the brain.
> > Ref: Cadena, Santiago A., et al. "How well do deep neural networks trained on object recognition characterize the mouse visual system?." (2019).
> > - Further, the same layer can also map onto multiple brain regions, thereby hiding meaningful differences between the neural representations in these regions. For e.g., when fitting the task-optimized model trained on MS-COCO (ResNet-50 MS-COCO) to brain response data, we found that all the 5 visual ROIs achieved best accuracy with the same layer (last layer of the res2 block).

---

> > > ### Comment · Reviewer_xZPs · 2022-08-04
> > > **Thank you for the revision and the clarifications!**
> > >
> > > This version of the manuscript is greatly improved so I will go ahead and increase my score. I think this will be a solid contribution to the field.

---

### Official Review · Reviewer_3Hjp · 2022-07-09

**Rating:** 7
**Confidence:** 5
**Soundness:** 4 excellent
**Presentation:** 3 good
**Contribution:** 3 good

**Summary:**

This work builds end-to-end voxel-level encoding models for fMRI recordings of a popular publicly available dataset of natural scene images (NSD). The authors perform extensive comparisons against networks optimized for image classification on different types of data, and show that their proposed response-optimized networks are better able to generalize to heldout subjects, are more sample-efficient, and better generalize to out of distribution images. The work also offers several interpretability analysis of the representations learned by the response-optimized networks and shows that they appear to align with human similarity judgments and have more shape bias than a task-optimized network.

**Questions:**

Major:
1. I would like to hear the author’s response to the point I raised under Weaknesses about a fairer comparison to a task-optimized model that is actually trained on the NSD training images.
2. Please discuss previous related work that has incorporated brain responses during training of models (see under Weaknesses for a few that I am aware of)
3. The paper contains a mix of encoding model results and RSA. Can the authors comment on why this was done, as opposed to only using encoding/decoding analyses?
4. Can the authors clarify how many images were used for train, validation, and test? L112 says that 1k were used for test and 37k for train and validation, but there are 37k images in total.

Minor:
- Related work to the finding that predictive models can denoise brain data: Ravishankar et al. 2021, Frontiers Comp Neuro showed that predictive models can help with improving both encoding and decoding performance (for single-trial MEG) https://www.frontiersin.org/articles/10.3389/fncom.2021.737324/full
- Please include a 1 sentence description in the main paper of the type of noise ceiling computation that was done.
- Lots of details relegated to supplementary with the text (see Suppl. Methods). The supplementary is long so please include specific pointers to where in the supplementary this information can be found.
- Calling the object category vector “semantic” is strange because all of the model representations contain semantic information. I would change the name to something more descriptive. Also another term to describe this “binary vector” is a k-hot encoding.
- Clarify whether the subjects were in a common space (e.g. MNI).
- The supplementary can benefit from proofreading (I spotted several typos from a quick look)

Suggestions for future/follow-up work:
- The PC interpretation in Fig 5D is nice, but it would be good to do something a bit more principled. If you define a set of semantic norms for each THINGS image (those can be sampled from already available datasets), you may be able to actually show what semantic dimensions are highly associated with each ROI.


**Limitations:**

This work directly integrates brain recording measurements into the training of deep learning models. It should discuss any possible undesirable biases that can arise from this (for instance, any additional societal bias).

**Strengths And Weaknesses:**

Strengths:

This work is very rich. It has a good technical contribution and the scientific interpretation is top-notch. I found the following highlights about the proposed method especially important:
- Good generalization to held-out subjects
- Good generalization to out of distribution images
- Increased shape-bias
- An improved sample efficiency

Weakness:

- No discussion of related work, this works needs to be addressed in order for this work to be published. Works that specifically integrate brain recordings during the training of models are important to mention and discuss with respect to the current work. Here are a few that I'm familiar with, and some of them even use the same image dataset:
    - Seeliger et al. 2021, PLoS Comp Bio https://journals.plos.org/ploscompbiol/article?id=10.1371/journal.pcbi.1008558
    - St-Yves et al. 2022, bioRxiv https://www.biorxiv.org/content/10.1101/2022.01.21.477293v1
    - Khosla and Wehbe, 2022 bioRXiv https://www.biorxiv.org/content/10.1101/2022.03.16.484578v1
    - In language: Schwartz et al. 2019 NeurIPS: https://proceedings.neurips.cc/paper/2019/hash/2b8501af7b64d1aaae7dd832805f0709-Abstract.html

- Using an AlexNet that is trained on ImageNet as that task-optimized model baseline doesn’t seem like the best choice. The NDS images are from the Coco dataset which has a different distribution from ImageNet, and therefore the test NSD images are much more out of domain for the task-optimized model than they are for the response-optimized model. A more fair comparison as a task-optimized model is to either train the same CNN architecture (minus the linear layer on top) to classify the NDS images, or at least fine-tune AlexNet on the training NDS images. Similarly, in the generalization to ImageNet experiments, the authors compare to a baseline model that has been randomly initialized. A stronger (and fairer) baseline here would be a task-optimized model that has the same architecture as the response-optimized and was trained with the same NSD images, but with a different objective (image classification, rather than brain prediction).

- The organization can be improved. Currently the Materials and Methods section is very long (ends on page 5) and details all experiments. By the time that the corresponding results come around, the reader has forgotten the necessary details from the experiment. I suggest describing the materials and main models in this section, and then moving the description of each analysis to the corresponding place in the results.

---

> ### Author Response · Authors · 2022-08-02
> **Response to Reviewer 3Hjp [Part 1]**
>
> Thank you for your positive comments and actionable suggestions. Please find our comments below.
>
> ### Missing literature
> We have added a more extensive literature review and better situated our work in the context of the relevant literature.
> “Studies have already revealed remarkable inconsistencies between these models and human behavior and gaps remain in their respective robustness to distortions and image-level consistency etc (Geirhos et al., 2021). Recently, the idea of using neural data in the model development process has gained some traction, and demonstrated success in both increasing model-human behavior alignment (Federer et al., 2020, Safarani et al., 2021, Dapello et al., 2022) and increasing the model-brain alignment while revealing new insights about information processing in neural systems (Schwartz et al. 2019, Seeliger et al., 2021, Khosla & Wehbe, 2022, St. Yves et al., 2022).”
>
> ### Suggestion about stronger baselines
> Thank you for these very useful suggestions about baselines.
>
> (i) We have now included several new competitive baselines in our quantitative results for prediction accuracy to enable a better comparison with the state-of-the-art in the field. These include more recent SOTA task-optimized models, including model architectures like ResNet-50, CORnet-S and DenseNet optimized for image recognition on the ImageNet database. Since these models can, in principle, demonstrate lower prediction performance due to the domain shift between ImageNet and NSD stimuli (as noted by the reviewer), we also employed a ResNet-50 architecture pre-trained for object detection on the MS-COCO dataset as an additional baseline (as NSD stimuli were drawn from the MS-COCO dataset). Details of these models have been listed in Appendix Table A.2 (also shown below). The results confirm that the proposed response-optimized model performs competitively with these strong baselines. We note that the MS-COCO-trained task-optimized model performs poorly when compared to the ImageNet-trained models. We speculate this might be because ImageNet is a richer dataset with an order of magnitude more stimuli and richer labels (1,000 categories v/s 80 object categories in the MS-COCO dataset) than MS-COCO. On the NSDsynthetic dataset, we observe that the ResNet-50 task-optimized model performs better than the response-optimized model in region VO; we have toned down our claims accordingly.
>
> (ii) We have also included an additional competitive baseline for the shape bias results. Here, we train a baseline DNN for multi-label object classification on the entire MS-COCO dataset, based on the reviewer’s suggestion. Since NSD images were drawn from the MS-COCO dataset, this serves as a useful and strong control for assessing features that may arise simply due to the dataset distribution. Importantly, the model employs the same backbone architecture as the proposed rotation-equivariant response-optimized model. Assessing the shape/texture bias (on the texture-shape cue conflict) and transfer performance of this model on the Silhouettes dataset reveal its strong bias for texture, in stark contrast to response-optimized models trained with stimuli from the same data distribution.
>
> ### Organization of the content
> Thank you for the suggestion. We have moved the Methods for some individual analyses into the Results section for continuity and clarity. We have also added relevant pointers to the sections in Appendix wherever applicable. Based on Reviewer #1’s suggestions, we have also added a summary table (Appendix Table A.1) describing all the analysis methods and baselines employed in this study.

---

> > ### Author Response · Authors · 2022-08-02
> > **Response to Reviewer 3Hjp [Part 2]**
> >
> > ### Incoherent baselines
> >
> > We have summarized  all evaluation methods and baselines used in the study in Appendix Table A.1. Our choice of methods was driven by several factors, including the dominance of the method in prior literature, its appropriateness for the research question, and its ease of evaluation.
> >
> > For evaluation against neural data, we consistently used Pearson’s R between the predicted and measured responses. For generalization to novel fMRI datasets, we used RSA instead of neural predictivity since some of the novel datasets employed in this study (e.g. Cichy et al., 2019) only made a public release of the RDMs, without releasing the raw voxel responses. Further, the goal of this latter analysis was to characterize whether the models indeed captured features idiosyncratic to their respective ROIs (in a dataset-neutral way). Thus, we only compared response-optimized models of different visual regions among themselves (and not task-optimized networks) for this analysis.
> >
> > For comparison with behavioral data, we favor the RSA approach over a linear decoding approach since (a) representational geometry provides a general characterization of the neural code, (ii) RSA enables us to test the representational feature space as is (without any fitting), making it easier to compare different models without the risk of overfitting. It has been widely used for assessing the informational content in neural representations. While linear decoder/mixed RSA methods can be useful since they allow us to account for differing prominence or gain of different features in the representation, RSA is a stricter measure. Even without any fitting, we found that the representations in models of higher order brain areas exhibit remarkable similarity with human perception (‘nsdmeadows’) and demonstrate a higher separability of object category information.
> >
> > ### Number of images for training and evaluation.
> > We apologize for confusing wording. We meant to say that the first group of NSD participants saw 37K images altogether. We have fixed this in the revised manuscript.
> >
> > We further incorporated all the suggestions listed by the reviewer in the 'Minor Comments' section. Thank you for the careful reading of our manuscript. These comments have improved our manuscript greatly.

---

> > > ### Comment · Reviewer_3Hjp · 2022-08-05
> > > **thanks for the additional analyses**
> > >
> > > Thanks for the thorough response and including additional baselines. My questions on that front are addressed.
> > >
> > > One follow-up: since completing the review, I have read the papers by Khosla and Wehbe, 2022 bioRXiv https://www.biorxiv.org/content/10.1101/2022.03.16.484578v1 and St-Yves et al. 2022, bioRxiv https://www.biorxiv.org/content/10.1101/2022.01.21.477293v1 more carefully and I believe they are both extremely related to the current work (training end-to-end encoding models, NSD dataset, comparisons with task-optimized networks). I would like the authors to address these two specific works in more detail and dis cuss the difference in contributions between the current work and these works. I understand that these works were made available recently (one in Jan and the other in March), and the authors should feel free to call them concurrent with the current work but I do believe they deserve a more in-depth discussion for the benefit of the reader.

---

> > > > ### Author Response · Authors · 2022-08-07
> > > > **Response to the follow-up [3Hjp]**
> > > >
> > > > Thank you again for suggesting those baselines - they have considerably improved our manuscript.
> > > > Thank you for the suggestion in the follow-up. We agree that these works are very relevant to the present study and deserve a more detailed treatment in our paper. We have added a short line in the discussion (main text) and provided a more in-depth comparison in Appendix C.1. We reproduce the text from the revised manuscript here for convenience:
> > > >
> > > > "Concurrent with the present work, recent papers (St. Yves et al., 2022, Khosla & Wehbe, 2022) have demonstrated that when trained with large amounts of data such as NSD, the response-optimization approach can perform competitively with state-of-the-art task-optimized models. Khosla et al. adopt similar neural network architectures and compare asymptotic performance and sample complexity in models of specialized category-selective regions of the brain, such as FFA and EBA, and do not study the ventral visual cortical hierarchy (the ROIs investigated in this study). St. Yves et al. study the early to intermediate visual cortex (V1-V4), specifically the role played by the brain hierarchy in training brain-optimized models that are not hierarchical by design, presenting alternate evidence against the hierarchical nature of neural representations. Our work differs from these studies in several respects: (i) We develop models for regions along the *entire* ventral visual hierarchy, including higher-order ROIs like LO and VO (the human analogues of IT). (ii) We study the generalization of response-optimized models to datasets beyond NSD and compare against a richer battery of task-optimized models. (iii) We demonstrate the denoising capability of models, showing that the predicted activity not only serves as a useful surrogate for the measured activity, but also yields a better match to neural data in other datasets and human behavior. (iv) While St. Yves et al. also demonstrate the ability of response-optimized models to recover the retinotopic organization in early visual cortex, they do so at the qualitative level of capturing size-eccentricity relationships. In the present study, we perform rigorous quantitative comparisons and sample complexity analysis on the ability of response-optimized models to capture the precise spatial receptive field tuning of individual voxels. (v) St. Yves et al. focus on network architectures and ask if hierarchy, at the level of model architecture, is needed to explain brain activity. Here, on the other hand, we ask: what *features* emerge spontaneously in networks optimized to explain brain activity and what kinds of biases do these networks exhibit (shape vs. texture)? We demonstrate increasing separability of categorical information and alignment with human perception in the optimized *models* of ventral visual visual stream regions, providing a model abstraction of this key neural phenomenon. In the future, we hope these can be richly interrogated to generate mechanistic hypotheses. The strong shape bias in response-optimized, but not task-optimized models, further has interesting implications for computer vision models. Altogether, these studies and our work invite a shift from the task-optimized modeling framework, providing an alternate modeling strategy to understand neural representations."

---

> > > > > ### Comment · Reviewer_3Hjp · 2022-08-07
> > > > > **Follow up addessed**
> > > > >
> > > > > Thank you, this added discussion addresses my follow-up question.

---

### Official Review · Reviewer_xc4V · 2022-07-10

**Rating:** 7
**Confidence:** 5
**Soundness:** 3 good
**Presentation:** 2 fair
**Contribution:** 3 good

**Summary:**

The paper proposed and encoding model using large fMRI datasets to predict the brain response to visual images in different regions of the ventral stream of the visual cortex. They test the model with different paradigms such as: generalization to out-of-distribution images, generalization to other subjects, spatial tuning in the early visual areas, separability of objects in higher visual areas, alignment with human perception, and shape bias. They then visualized the preferred images by different voxels predicted by the model. The main novelty here are the shape bias and the ability of the model to achieve these above things concurrently. Each one of these was observed already in previous models but not together. The paper is quite dense and the authors attempted their best to fit within the page limit.

**Questions:**

- Do you have any explanation for the aggregated distribution of points in figure 4A which seems to be relatively ordinal in comparison to the predicted?



**Limitations:**

- Encoding models are inherently limited by their dependence on the images solely unless other mental or prior information are added
- Testing on a new dataset limits their ability to test the model against other models that focus on classification.
- The appeal of encoding models is to understand how the model is able to achieve these feats that are similar to human visual cortex which is not addressed here.

**Strengths And Weaknesses:**

Strengths:
- The paper succeeds in having a strong prediction accuracy for OOD images. While, many other models have managed to do that, it is not an easy task.
- Shape bias is one novelty over the traditional DNN models for CV.
- The model shows multiple similarities with the ventral stream properties instead of focusing on one or two.

Weaknesses:
- The paper is quite dense leading the authors to omit many necessary information and to have figures too small and cluttered.
- Literature review in this field is lacking many important relevant references that discuss the idea of the similarity of hierarchy between humans and DNNs. To name a few:
    - Horikawa, Tomoyasu, and Yukiyasu Kamitani. "Generic decoding of seen and imagined objects using hierarchical visual features." Nature communications 8.1 (2017): 1-15.
    - Abdelhack, Mohamed, and Yukiyasu Kamitani. "Sharpening of hierarchical visual feature representations of blurred images." Eneuro 5.3 (2018).
   - Tsunoda, Kazushige, et al. "Complex objects are represented in macaque inferotemporal cortex by the combination of feature columns." Nature neuroscience 4.8 (2001): 832-838.
   - Schrimpf, Martin, et al. "Brain-score: Which artificial neural network for object recognition is most brain-like?." BioRxiv (2020): 407007.
   - Schrimpf, Martin, et al. "Integrative benchmarking to advance neurally mechanistic models of human intelligence." Neuron 108.3 (2020): 413-423.
   - Federer, Callie, et al. "Improved object recognition using neural networks trained to mimic the brain’s statistical properties." Neural Networks 131 (2020): 103-114.


- The paper tests many different conditions with their model leading the paper to be increasingly complex since each condition is done via a separate analysis. One suggestion is to have a summary table of test and how it is measured and baselines so one can understand the big picture.

- Each of the baseline model is trained very differently which makes them not really good baselines

- Viewing figure 6A does not support the statement in the text that preferred stimuli increase in complexity

---

> ### Author Response · Authors · 2022-08-02
> **Response to Reviewer xc4V [Part1]**
>
> We thank the reviewer for their thoughtful comments and actionable suggestions. We address their concerns below:
>
> ### Densely packed content
> We realize this paper is quite dense.  We tried to make our presentation as clear as possible while adhering strictly to the page limit, but we could have missed the mark in certain places.  We have worked significantly on our presentation in the revised version. This includes the following changes:
> (i) We have created a new table in the Appendix (Table A.1) describing all the analyses, evaluation metrics and baselines.
> (ii) We have judiciously moved some content to the Appendix without affecting the narrative of the text. This enabled us to further clarify the Methods in the text. We have moved the Methods for some individual analyses into the Results section for continuity and clarity.
> (iii) We have updated Figure 6 to remove the clutter and show only the most significant shape bias results in the revised version.
> (iv) We have revised Figure 2 & 3 and improved their presentation.
> (v) We have added pointers to the relevant subsections in the Appendix for each analysis.
> We believe that each of the experiments conducted in the paper provide a novel perspective on the strengths of the proposed response-optimized model, and we think our paper benefits significantly from each result.
>
> ### Missing literature
> We thank the reviewer for pointing out the missing literature. We have now included a much more extensive literature review in the introduction of the revised manuscript.
>
> “These carefully crafted experiments, for instance, revealed the presence of edge detectors in the primary visual cortex (Hubel and Weisel, 1962) and sparked the search for the `optimal input' beyond early visual areas across the visual cortical hierarchy. Significant efforts have also been made to understand the feature complexity of tuning in V4 and IT neurons (Desimone et al., 1984,, Tanaka et al., 1999, Pasupathy et al., 1999, Tsunoda et al., 2001).”
>
> “Recently, deep convolutional neural networks (CNNs) optimized for computer vision (CV) tasks, have emerged as the most accurate models of the primate ventral visual stream (Yamins et al., 2014, Schrimpf et al., 2020a, Schrimpf et al., 2020b, Storrs et al., 2020). Furthermore, these CV task-optimized models offer a hierarchical correspondence to visual areas along the ventral stream: early CNN layers best match V1, while intermediate and late layers best correspond to V4 and IT (Khaligh-Razavi and Kriegeskorte 2014, Yamins et al., 2014, Guclu et al., 2015,  Cichy et al. 2016, Eickenberg et al., 2016, Horikawa and Kamitani, 2017, Wen et al., 2017, Abdelhack and Kamitani, 2018).”
>
> “Studies have already revealed significant  inconsistencies between these models and human behavior and gaps remain in their respective robustness to distortions and image-level consistency, etc (Geirhos et al., 2021). Recently, the idea of using neural data in the model development process has gained some traction, and demonstrated potential  for increasing both model-human(Federer et al., 2020, Safarani et al., 2021, Dapello et al., 2022) and model-brain alignment, while revealing new insights about information processing in neural systems (Schwartz et al. 2019, Seeliger et al., 2021, Khosla & Wehbe, 2022, St. Yves et al., 2022).”

---

> > ### Author Response · Authors · 2022-08-02
> > **Response to Reviewer xc4V [Part2]**
> >
> > ### Inadequate baselines
> > (i) We have now included several new competitive baselines in our quantitative results for prediction accuracy to enable a better comparison with the state-of-the-art in the field. These include more recent SOTA task-optimized models, including model architectures like ResNet-50, CORnet-S and DenseNet optimized for image recognition on the ImageNet database. Since these models can, in principle, demonstrate lower prediction performance due to the domain shift between ImageNet and NSD stimuli, we also employed a ResNet-50 architecture pre-trained for object detection on the MS-COCO dataset as an additional baseline (as NSD stimuli were drawn from the MS-COCO dataset). Details of these models have been listed in Appendix Table A.2. The results confirm that the proposed response-optimized model performs competitively with these strong baselines. On the NSDsynthetic dataset, we observe that the ResNet-50 task-optimized model performs better than the response-optimized model in region VO; we have toned down our claims accordingly.
> >
> > (ii) We have also included an additional competitive baseline for the shape bias results (Figure 6). Here, we train a baseline DNN for multi-label object classification on the entire MS-COCO dataset. Since NSD images were drawn from the MS-COCO dataset, this serves as a useful and strong control for assessing features that may arise simply due to the dataset distribution. Importantly, the model employs the same backbone architecture as the proposed rotation-equivariant response-optimized model. Assessing the shape/texture bias (on the texture-shape cue conflict) and transfer performance of this model on the Silhouettes dataset reveal its strong bias for texture, in stark contrast to response-optimized models trained with stimuli from the same data distribution.
> >
> > ### Incoherent baselines
> >
> > We have reported all evaluation methods and baselines used in the study in Appendix Table A.1. Our choice of analyses methods was driven by several factors, including the dominance of the method in prior literature, its sensibility for the research question, and its ease of evaluation.
> >
> > For evaluation against neural data, we consistently used Pearson’s R between the predicted and measured responses. For generalization to novel fMRI datasets, we used RSA instead of neural predictivity since some of the novel datasets employed in this study (e.g. Cichy et al., 2019) only made a public release of the RDMs, without releasing the raw voxel responses. Further, the goal of this latter analysis was to characterize whether the models indeed captured features idiosyncratic to their respective ROIs (in a dataset-neutral way). Thus, we only compared response-optimized models of different visual regions among themselves (and not task-optimized networks) for this analysis.
> >
> > For comparison with behavioral data, we favor the RSA approach over a linear decoding approach since (ia) representational geometry provides a general characterization of the neural code, (ii) RSA enables us to test the representational feature space as is (without any fitting), making it easier to compare different models without the risk of overfitting, and (iii). It has been widely used for assessing the informational content in neural representations. While linear decoder/mixed RSA methods can be useful since they allow us to account for differing prominence or gain of different features in the representation, RSA is a stricter measure. Even without any fitting, we found that the representations in models of higher order brain areas exhibit remarkable similarity with human perception (‘nsdmeadows’) and demonstrate a higher separability of object category information.
> >
> > For the shape bias analysis, we used the standard evaluation protocol of Geirhos et al. (2018), who first demonstrated the strong texture bias of CV task-optimized CNNs.

---

> > > ### Author Response · Authors · 2022-08-02
> > > **Response to Reviewer xc4V [Part3]**
> > >
> > > ### Viewing figure 6A does not support the statement in the text that preferred stimuli increase in complexity
> > > We address this concern with two further analyses:
> > > (i) We measured the complexity of optimized images using a measure based on compression ratios, that was previously adopted to quantify complexity of synthesized images (Rose et al., 2021). Briefly, this metric, based on the discrete cosine transform, estimates the minimum number of coefficients (corresponding to different spatial frequencies) needed to accurately reconstruct the image, which in turn indicates the ‘compressibility’ (or inversely, complexity) of an image. We quantified the complexity of all images in Figure 6A using this metric and found that the complexity does indeed increase along the ventral visual hierarchy (Results are added as Appendix Table A.3). We do, however, note that this metric is limited and speculative, as complexity can be defined in several ways. Future work needs to be done to more rigorously characterize these preferential patterns.
> > >
> > > | Model ROI      | Complexity |
> > > | :---        |    :----:   |
> > > | V1      | 0.439+/-0.041       |
> > > | V2   | 0.433+/-0.012        |
> > > | V4   | 0.449+/-0.034       |
> > > | LO   | 0.536+/-0.079        |
> > > | VO   | 0.637+/-0.012        |
> > >
> > >
> > > (ii) We also extracted the top 3 natural images among the THINGS database (~27,000 images) that produce the highest predicted response for each V1 or VO voxel (the two ends of the hierarchy) visualized in Figure. 6A. The natural images contain similar featural configurations as their corresponding synthesized counterparts: the natural images for V1 voxels contain objects at specific orientations or high frequency spatial patterns reminiscent of the orientation preferences visible in the synthesizes images, whereas the images for VO contain complex shapes like concentric circles, rectangles and hexagons, again in agreement with their respective synthesized images. We hope that these results, shown as Appendix Figure A.7,  provides further evidence that the patterns in Fig.6A are more complex for higher-order regions like VO than the primary visual cortex.

---

> > > > ### Author Response · Authors · 2022-08-02
> > > > **Response to Reviewer xc4V [Part4]**
> > > >
> > > > ### Question about Figure 4A
> > > > We are afraid we’re not sure about the meaning of this question. If the reviewer is referring to the fact that the measured eccentricity values appear quantized, we think it’s because the pRF data released with NSD was given in degrees of visual angle (integers). In case the reviewer is referring to discrepancies between model-predicted and localizer-estimated pRF parameters, we think there are several factors that could influence this. First, we impose no constraints (except positivity) on the spatial readout weights and do not impose any 2D Gaussian structure during model training. This could, in principle, lead to spatially distributed weights and erroneous calculations of pRF eccentricity and polar angle when the weights don’t obey a Gaussian structure. In practice, however, the visualization of optimized spatial weights corresponding to randomly selected voxels suggested that they do have a localized receptive field structure. Discrepancies could also arise due to the noise in pRFestimation procedure, particularly since the number of artificial stimuli employed in the pRF localizer scans is relatively low. In fact, the pRF parameters estimated from our encoding model are spatially smoother (despite no smoothness constraints), suggesting that they could potentially be more accurate than the localizer-estimate retinotopy. The differences might also reflect differential processing of naturalistic and artificial stimuli.
> > > >
> > > > Finally, thank you for giving us the opportunity to strengthen our manuscript with your valuable comments.

---

> > > > > ### Comment · Reviewer_xc4V · 2022-08-03
> > > > > **Increased my score**
> > > > >
> > > > > Thank you very much for the responses and the new version. It is so much improved and the supplements add a lot of details. I have increased my score for it.
> > > > >
> > > > > It would also be very good if you could address the last point of the limitations in the discussion. You already had a good start at the end of the discussion but maybe a bit more detail would make it much more informative as to what would be the next steps. Maybe you could discuss points such as what the structure of the model reveals about brain connections and how it can be improved to include more biological relevance.

---

> > > > > > ### Author Response · Authors · 2022-08-04
> > > > > > **Nice suggestion; added that in the discussion**
> > > > > >
> > > > > > Thank you so much for your careful reading of our response and the revised manuscript.
> > > > > > We agree that how the human brain achieves such a feat is the ultimate question and we have now added a brief discussion on this limitation/future direction at the end of our revised manuscript.
> > > > > > Thank you for the suggestion.

---

### Meta-Review · Area_Chair_Ktju · 2022-08-26

**Recommendation:** Accept
**Confidence:** Certain

**Metareview:**

The authors leverage a large fMRI dataset to fit  “hypothesis-agnostic”/data-driven encoding models to brain data (as opposed to task-optimized ones -- as done in most of the related work). While there is general agreement that this is not a groundbreaking advance (there is already similar work published), the reviewers agreed that the paper was technically strong (good generalization to held-out subjects, good generalization to OOD images,  increased shape-bias, improved sample efficiency). The interpretation of results for neuroscience was rated as "top-notch".  The paper received a rare unanimous clear accept. The AC recommend acceptance.

**Award:**

No

---

### Decision · Program_Chairs · 2022-09-14

Accept